

# Long-term trends in total inorganic nitrogen and sulfur deposition in the U.S. from 1990 to 2010

Yuqiang Zhang[1], Rohit Mathur[2], Jesse O. Bash[2], Christian Hogrefe[2], Jia Xing[3], Shawn J. Roselle[2]

[1]Oak Ridge Institute for Science and Education (ORISE) Fellowship Participant at US Environmental Protection Agency, Research Triangle Park, NC 27711, USA

[2]US Environmental Protection Agency, Research Triangle Park, NC 27711, USA

[3]School of Environment, Tsinghua University, Beijing, 100084, China

*Correspondence to*: Yuqiang Zhang (zhang.yuqiang@epa.gov; yuqiangzhang.thu@gmail.com)

**Abstract.** Excess deposition (including both wet and dry deposition) of nitrogen and sulfur are detrimental to ecosystems. Recent studies have investigated the spatial patterns and temporal trends of nitrogen and sulfur wet deposition, but few studies have focused on dry deposition due to the scarcity of dry deposition measurements. Here, we use long-term model simulations from the coupled WRF-CMAQ model covering the period from 1990 to 2010 to study changes in spatial distribution as well as temporal trends in total (TDEP), wet (WDEP) and dry deposition (DDEP) of total inorganic nitrogen (TIN) and sulfur ($TSO_4$). We first evaluate the model's performance in simulating WDEP over the U.S. by comparing the model results with observational data from the U.S. National Atmospheric Deposition Program. The coupled model generally underestimates the WDEP of both TIN (including both the oxidized nitrogen deposition-$TNO_3$, and the reduced nitrogen deposition-$NH_X$) and $TSO_4$, with better performance in the eastern U.S. than the western U.S. TDEP of both TIN and $TSO_4$ show significant decreases over the U.S., especially in the east due to the large emission reductions that occurred in that region. The decreasing trends of TIN TDEP are caused by decrease of $TNO_3$, and the increasing trends of TIN deposition over the Great Plains and Tropical Wet Forests regions are caused by increases in $NH_3$ emissions although it should be noted that these increasing trends are not significant. TIN WDEP shows decreasing trends throughout the U.S., except for the Marine West Coast Forest region. TIN DDEP shows significant decreasing trends in the region of Eastern Temperate Forests, Northern Forests, Mediterranean California and Marine West Coast Forest, and significant increasing trends in the region of Tropical Wet Forests, Great Plains and Southern Semi-arid Highlands. For the other three regions (North American Deserts, Temperate Sierras and Northwestern Forested Mountains), the decreasing or increasing trends were not significant. Both the WDEP and DDEP of $TSO_x$ have decreases across the U.S., with a larger decreasing trend in the DDEP than that in the WDEP. Across the U.S. during the 1990-2010 period, DDEP of TIN accounted for 58-65% of TDEP of TIN. TDEP of TIN over the U.S. was dominated by deposition of $TNO_3$ during the first decade, which then shifts to reduced nitrogen ($NH_X$) dominance after 2003 resulting from combination





of $NO_x$ emission reductions and $NH_3$ emission increases. The sulfur DDEP is usually higher than the sulfur WDEP until recent years, as the sulfur DDEP has a larger decreasing trend than WDEP.

# 1 Introduction

Increased nitrogen and sulfur deposition is detrimental to ecosystems, since it leads to decreased biological diversity (Clark and Tilman, 2008; Clark et al., 2013; Stevens et al., 2004), increased terrestrial and aquatic eutrophication and acidification (Bouwman et al., 2002; Bowman et al., 2008; Fisher et al., 2011; Greaver et al., 2012; Savva and Berninger, 2010). The primary sources for nitrogen deposition are nitrogen oxides ($NO_x \equiv NO + NO_2$) and ammonia ($NH_3$), which both have anthropogenic and natural sources. The major source for $NO_x$ is from the combustion of fossil fuels in industry and energy use (Elliott et al., 2007; Lamarque et al., 2010). For $NH_3$, 80% of the total emissions are from livestock manure management and chemical fertilizer in 2005 as estimated from the U.S. National Emission Inventory (Reis et al., 2009), which are not regulated under current legislation and underwent significant increases over the past decades (Xing et al., 2013; Warner et al., 2017). Another possible source of $NH_3$ emissions are from vehicles which may be twice higher than the emission estimates in the current NEI (Sun et al., 2016). The primary emission source for the sulfur deposition is sulfur dioxide ($SO_2$) which also mainly originates form fossil-fuel combustion (Smith et al., 2011).

The ultimate fate for $NO_x$, $NH_3$ and $SO_2$ is removal by wet scavenging and uptake by terrestrial and aquatic ecosystems (Greaver et al., 2012). Wet deposition (WDEP), in the form of rain or snow, is relatively easy to measure. Several observation networks were established to provide reliable long-term records of WDEP, such as the European Monitoring and Evaluation Programme (EMEP) in Europe, the National Acid Deposition Monitoring Network (NADMN) in China, the Canadian Air and Precipitation Monitoring Network (CAPMoN) in Canada, and the National Atmospheric Deposition Program's National Trends Network (NADP/NTN) in the U.S. (Xu et al., 2015). These data have been extensively used to quantity the sources, pattern, and temporal trends of WDEP of major species worldwide (EEA 2011; Jia et al., 2014; Cheng and Zhang, 2017; Lajtha and Jones, 2013; Du et al., 2014; Sickles II and Shadwick, 2007a, 2007b, 2015, to name a few). However, the majority of these studies discussed WDEP based on the measurements only, and neglected the discussion of the spatial distribution and trends of dry deposition (DDEP), as no direct DDEP measurements are available at these networks. The calculated values at some sites, such as for the Clean Air Status and Trends Network (CASTNET) and CAPMoN, however cannot be easily spatially interpolated due to their complexity (Schwede and Lear, 2014). DDEP can contribute up to two-thirds of total deposition (TDEP) of nitrogen, and neglecting it can lead to substantial underestimation of the total flux (Flechard et al., 2011; Vet et al.,

2014). Also, accurate estimates of TDEP are usually required to assess the impacts of excess nitrogen and sulfur deposition on ecosystem health, such as critical load exceedances and species loss. (Simkin et al., 2016).

To address these challenges, global and regional chemical transport models (CTMs) have been extensively used in recent years to quantify the sources and distribution of both WDEP and DDEP (Mathur and Dennis, 2003; Galloway et al., 2008; Paulot et

5 al., 2013; Sanderson et al., 2008; Zhang et al., 2012; Zhao et al., 2009; Zhao et al., 2015, 2017), to study the projected deposition changes in the future (Dentener et al., 2006; Larmarque et al., 2013; Ellis et al., 2013; Kanakidou et al., 2006; Sun et al., 2017), and also its effect on ecosystems (Simkin et al., 2016). CTMs can link the sources to the deposition through atmospheric chemistry and transport process, and can provide insights on the trends of TDEP and its components. In this study we quantify the long-term geographical patterns and temporal trends of TDEP, WDEP, and DDEP of total inorganic nitrogen and sulfur

10 over the continental U.S. based on a 21-year model simulation from 1990 to 2010 at 36km×36km. The paper is organized as follows. Section 2 describes the model configuration and observation datasets as used for model evaluation. The model evaluation results and the patterns and trends of inorganic nitrogen and sulfur deposition are presented in Section 3, followed by the conclusions in Section 4.

## 2 Methods

15 ### 2.1 Model setup

The long term simulations were previously performed using the coupled WRF-CMAQ model (Wong et al., 2012) with WRFv3.4 coupled with CMAQv5.02 driven by internally consistent U.S. emission inventories (Xing et al., 2013) covering the Continental U.S. (CONUS) domain discretized with a grid of 36 km horizontal resolution. Spatial and time varying chemical lateral boundary conditions were provided by the hemispheric WRF-CMAQ (Mathur et al., 2017) running over the same period

20 (Xing et al., 2015). Interested readers are referred to Gan et al., (2015, 2016) for detailed description of the settings of the CMAQ model and physical configurations of the WRF model (Table S1). The performance of the coupled WRF-CMAQ model for major trace gases, aerosol species and meteorological variables such as $O_3$, $PM_{2.5}$ and aerosol optical depth at both the hemispheric and regional scale have been extensively evaluated in previous studies (Xing et al., 2015a, b; Mathur et al., 2017; Gan et al., 2015, 2016; Astitha et al., 2017), and was shown skill in simulating the magnitudes and long-term trends of these

25 variables.



## 2.2 Wet deposition observations in the U.S.

A previous study using the offline CMAQ model has demonstrated moderate skill simulating WDEP from 2002 to 2006 (Appel et al., 2011). Here we evaluate the coupled WRF-CMAQ model's ability to simulate WDEP of nitrate ($TNO_3$), ammonium ($NH_X$) and sulfate ($TSO_X$) during 1990 – 2010 over the U.S., including both the interannual variability as well as long-term

trends. This is accomplished by comparing the model results with observations from the U.S. National Atmospheric Deposition Program (NADP, http://nadp.sws.uiuc.edu/ntn/), which measures total weekly wet deposition of these species. We first pair the wet deposition data between the observation and the model results in time and space, and then extract the annual deposition for the sites matching our criteria (at least 18 available years with 75% annual coverage for each year). Model data during periods of missing observations were not considered in either the statistical evaluation or the trends analysis. By applying the

criteria, we use information at 170 of 359 sites, with 141 sites in the eastern U.S. (east of 110˚W longitude) and 29 sites in the western U.S. (west of 110˚W longitude). The detailed site information and the number of years of observation data used for model evaluation can be found in supporting Table S2. In pairing the observed and modeled $TNO_3$ WDEP values (which combine WDEP of $NO_3^-$ and $HNO_3$), we multiply the model estimated $HNO_3$ WDEP with 0.984 to account for the transformation of $HNO_3$ to $NO_3^-$ in solution in the measurements. In pairing the observed and modeled $NH_X$ WDEP values

(which combine WDEP of $NH_4^+$ and $NH_3$), we multiply the model estimated $NH_3$ WDEP with 1.06 to account for the transformation of $NH_3$ to $NH_4^+$ in the rainwater in the measurements. In pairing the observed and modeled $TSO_X$ WDEP values (which combine WDEP of $SO_4^{2-}$ and $SO_2$), we multiply the model estimated $SO_2$ WDEP with 1.50 to account for the fact that $SO_2$ will be fully oxidized into $SO_4^{2-}$ during sampling (Appel et al., 2011).

For the model evaluation, we examine the correlation coefficients (R), Mean Bias (MB) as well as the normalized mean bias

(NMB):

$$NMB = \frac{\sum_1^N (Model - Obs)}{\sum_1^N Obs} \quad (1)$$

When discussing the model evaluation and deposition trends, we divide the U.S. into 10 ecological regions, following the North America Level I ecoregion definition (https://www.epa.gov/eco-research/ecoregions-north-america, accessed 08/01/2017), including Northern Forests, Northwestern Forested Mountains, Marine West Coast Forest, Eastern Temperate

Forests, Great Plains, North American Deserts, Mediterranean California, Southern Semi-arid Highlands, Temperate Sierras, and Tropical Wet Forests (supporting Fig. S1).



Errors in the simulated meteorology and precipitation in particular, can lead to errors in estimating WDEP in the CMAQ model. We follow the previous approach of Appel et al. (2011) to account for biases in modelled precipitation by adjusting the modelled WDEP as:

$$Bias\ Adjusted\ WD_{mod} = \frac{Precip_{Obs}}{Precip_{mod}} \times WD_{mod} \quad (2)$$

5 In equation 2., $WD_{mod}$ represents the WDEP from the model, $Precip_{Obs}$ represents annual or monthly accumulated observed precipitation, and $Precip_{mod}$ represents the corresponding annual or monthly accumulated precipitation from the model.

## 3 Results

### 3.1 Model evaluation for WDEP

After performing the annual precipitation adjustment for model simulated WDEP, we see that the correlation coefficients (R) 10 are slightly improved relative to using the unadjusted WDEP values (Table 1), increasing from 0.89 to 0.92 for TNO$_3$, from 0.77 to 0.81 for NH$_X$, and from 0.92 to 0.94 for TSO$_X$. There are no significant changes for R when we use the monthly precipitation adjustment compared with the annual precipitation adjustment. The model generally underestimates WDEP for both the eastern and western U.S., except for TSO$_X$ where the model tends to overestimate WDEP in the western U.S. (Figs. 1 and 2). The coupled WRF-CMAQ model generally overestimates the precipitation throughout U.S. (Fig. 2(d), supporting Fig. 15 S2), consistent with previous findings (Ran et al., 2015). After performing the precipitation adjustment, the NMB values increases for all the three species (Table 1). The model exhibit better performance for WDEP in east than that in west, considering both the R and the NMB, largely because of the complex terrain in the western U.S. (Appel et al., 2011)

The 21-yr average TNO$_3$ WDEP is highest in the Eastern Temperate Forest region, and lowest in the Southern Semi-arid Highlands, as seen from both observations and models (Table 2). The model generally underestimates the TNO$_3$ deposition for 20 all the regions with MB values ranging from -1.11 kg ha$^{-1}$ in the Southern Semi-arid Highlands to -3.73 kg ha$^{-1}$ in Tropical Wet Forests, except for the Marine West Coast Forest region where the model overestimates the TNO$_3$ WDEP, with MB values of 0.79 kg ha$^{-1}$. The correlation coefficients between the model and observations are generally much higher in the eastern U.S. (R larger than 0.80), than the western U.S. (R less than 0.70). The 21-yr average NH$_X$ WDEP is also highest in the Eastern Temperate Forest region, and lowest in the Southern Semi-arid Highlands (Table 3). The model generally underestimates the 25 NH$_X$ WDEP with MB values ranging from -0.26 kg ha$^{-1}$ yr$^{-1}$ in the Northwestern Forested Mountains to -0.81 kg ha$^{-1}$ in Tropical Wet Forests, and overestimates in the Marine West Coast Forest with MB of 0.24 kg ha$^{-1}$. The correlation coefficients between model and observations for NH$_X$ WDEP share similar spatial patterns with TNO$_3$ WDEP but have lower R values. The 21-yr



average TSO$_4$ deposition is highest in the Eastern Temperate Forests region, and lowest in the North American Desserts. Similar to TNO$_3$ and NH$_X$, the model underestimates the TSO$_4$ WDEP over most of the regions, but overestimates observed values in the Marine West Coast Forest and Mediterranean California. The R between the model and the observations are generally larger than 0.9 in the eastern U.S. but range from 0.46 to 0.79 in the western U.S.

Clear downward trends are seen for TNO$_3$ and TSO$_4$ WDEP from both the observations and model in Fig.2 (a, c), while NH$_X$ deposition exhibits much larger interannual fluctuations (Fig.2 (b)). From Fig. 3, we see much larger decreasing trends for TNO$_3$ and TSO$_4$ WDEP in the eastern U.S. than those in the western U.S. This is due to the fact that the emission reductions mostly occurred in the eastern U.S. (Xing et al., 2013) and the model captures this trend very well especially for TNO$_3$ and TSO$_4$ WDEP with R values of 0.94 and 0.95, respectively. A stronger decreasing trend over the Northern Forests and Eastern

Temperate Forests regions compared to other regions is observed for both TNO$_3$ and TSO$_4$ WDEP, and the model is also able to capture these very well but a slightly distinctions in trends for different ecoregions (Tables 2 and 4). We see that the model generally underestimates the decreasing WDEP trends for all the sites for TNO$_3$ and TSO$_4$ (Tables 2 and 4). For NH$_X$, we see increasing WDEP trends for most of the sites but the trends are not statistically significant (Table 3).

Compared with Appel et al. (2011), our results model results indicate larger bias for WDEP for both the eastern and western

U.S. (supporting Table 3). The NMB increasess for all the three species in our results from 2002 to 2006 after applying the precipitation-adjustment, which was also seen in Appel et al. (2011), except for TSO$_4$, which Appel et al. (2011) reported decreased bias after the precipitation adjustment. The discrepancies for the model performances between our study and Appel et al. (2011) could be caused by the grid resolutions, in which coarse resolution models (e.g. 36km in our study) are more challenging to simulate various chemical and physical processes compared with fine resolution (e.g. 12km used in Appel et

al., 2011).

### 3.2 Spatial patterns of modelled total deposition of nitrogen and sulfur

Table 5 shows that modeled TDEP of total inorganic nitrogen (TIN), i.e. the sum of TNO$_3$ and NH$_X$, is much higher in the Eastern Temperate Forests than any other ecoregion (regional average of 10.08 and 7.95 kg N ha$^{-1}$ in 1990 and 2010, respectively), followed by the Northern Forests and Mediterranean California regions. The hotspot for TIN TDEP has shifted

from the eastern U.S. in 1990 to the north central U.S. in 2010, with relative higher values in North Carolina (NC) and Pennsylvania (PA) (Fig. 4). During the period from 1990 to 2010, TIN TDEP has significantly decreased (with p <0.05 for the standard two-tailed Student's T-test) over several ecoregions, including Eastern Temperate Forests, Northern Forests, Mediterranean California and Marine West Coast Forest (decreasing trend of 0.12, 0.071, 0.038 and 0.017 kg N ha$^{-1}$ yr$^{-1}$




respectively). Slightly increasing but not statistically significant trends are estimated in TIN TDEP for the Great Plains and the Tropical Wet Forests while the remaining regions show statistically insignificant decreasing trends (Table 6). We see statistically significant increasing trends of TIN TDEP in eastern North Carolina (larger than 0.2 kg N ha$^{-1}$ yr$^{-1}$), which is mainly caused by the increase in NH$_X$ TDEP (Fig. 5) arising from increased NH$_3$ emission from hog farming (Xing et al., 2013; Paulot et al., 2014). There are also significant increasing trends of TIN TDEP over Iowa, Minnesota and South Dakota (larger than 0.04 kg N ha$^{-1}$ yr$^{-1}$) because of the increased NH$_X$ TDEP related to animal foster and corn plantation (Figs. 4 and 5). From Fig. 5, we see that the TIN TDEP decreasing trends predominantly result from the TNO$_3$ TDEP decreases across the U.S., with larger decreasing rates in the east than the west. The increasing TIN TDEP trends over the east and central states (such as North Carolina, Pennsylvania, and Virginia) were caused by the NH$_X$ TDEP increases which in turn arise from increases in NH$_3$ emissions (Paulot et al., 2013).

Similar to TIN TDEP, TDEP of total sulfur (TSO$_X$), i.e. the sum of SO$_2$ and SO$_4^{2-}$, shows a distinct spatial gradient from the east (usually larger than 9 kg S ha$^{-1}$) compared to the west (lower than 3 kg S ha$^{-1}$) (Fig. 4). In 1990, the TSO$_X$ was even higher than 30 kg S ha$^{-1}$ in some states of the central U.S., such as Indiana, Ohio, Pennsylvania, and West Virginia. In 2010, TSO$_X$ TDEP is still higher in the east than the west, but TSO$_X$ TDEP in the east has decreased by half (to lower than 15 kg S ha$^{-1}$) for most regions. From 1990 to 2010, the estimated TSO$_X$ TDEP exhibits significant trends across the U.S., with decreasing trends generally larger in the east (larger than 0.4 kg S ha$^{-1}$ yr$^{-1}$) and lower in the west (less than 0.2 kg S ha$^{-1}$ yr$^{-1}$) as a result of SO$_2$ decreases from the passage of the Clean Air Act Amendments of 1990. All the ecoregions experienced statistically significant decreases of TSO$_X$ TDEP over the past two decades, except for the Mediterranean California which showed and insignificant decreasing trend (Table 4). The largest decreasing trend was seen in the Eastern Temperate Forests region (-0.51 kg S ha$^{-1}$ yr$^{-1}$), followed by the Northern Forests (-0.23 kg S ha$^{-1}$ yr$^{-1}$) and the Great Plains (-0.082 kg S ha$^{-1}$ yr$^{-1}$).

### 3.3. Wet versus dry nitrogen and sulfur deposition trends in the U.S.

Fig. 6 shows that the TIN WDEP is higher in the east than the west, due to both greater precipitation (Fig. 2 (d)) and higher atmospheric burden of airborne reactive nitrogen in the east (Xing et al., 2013). In addition, estimated TIN WDEP shows widespread significant decreasing trends in the eastern U.S. while trends in the western U.S. generally have smaller magnitudes and often are not statistically significant. The most significant decreasing region is Eastern Temperate Forests, with an annual decrease of -0.070 kg N ha$^{-1}$ yr$^{-1}$, followed by Northern Forests (-0.037 kg N ha$^{-1}$ yr$^{-1}$) and Great Plains (-0.023 kg N ha$^{-1}$ yr$^{-1}$) (supporting Table S4). The decreasing trends of TIN WDEP was mainly caused by the WDEP of TNO$_3$ (supporting Fig. S4a, and Table S4). There are no significant changes for WDEP of NH$_X$ (supporting Fig. S4b), consistent with previous findings



(Lajtha and Jones, 2013). TIN DDEP is higher in the eastern U.S. and lower in the northwestern and central U.S. Significant decreasing trends for the TIN DDEP were seen over the Eastern Temperate Forests (-0.049 kg N ha$^{-1}$ yr$^{-1}$), Northern Forests (-0.033 kg N ha$^{-1}$ yr$^{-1}$), Mediterranean California (-0.032 kg N ha$^{-1}$ yr$^{-1}$), and Marine West Coast Forest regions (-0.022 kg N ha$^{-1}$ yr$^{-1}$) (supporting Table S5). The decreases of TIN DDEP over these regions were dominated by the DDEP of TNO$_3$

(supporting Fig. S4c, and Table S5). In contrast, there are significant increasing trends of TIN DDEP over the Tropical Wet Forests (0.027 kg N ha$^{-1}$ yr$^{-1}$), Great Plains (0.026 kg N ha$^{-1}$ yr$^{-1}$), and Southern Semi-arid Highlands (0.009 kg N ha$^{-1}$ yr$^{-1}$). These increases are caused by the DDEP of NH$_x$ (supporting Fig. S4d, and Table S5).

Fig. 7 shows distinct spatial distribution for both the WDEP and DDEP of sulfur, with much higher value in the eastern U.S. in vicinity and downwind of major source. Significant decreasing trends are noted for both the wet and dry TSO$_X$ deposition

for all the ecoregions, except for the Marine West Coast Forest and Mediterranean California where TSO$_X$ WDEP were estimated to increase, though the trend was not statistically significant (supporting Tables S4 and S5). TSO$_X$ DDEP trends were larger or comparable to TSO$_X$ WDEP trends for the majority of the regions, except for Southern Semi-arid Highlands, Temperate Sierras and Tropical Wet Forests where the magnitude of the decreasing trends for DDEP were lower than those for WDEP.

**3.4 Deposition budget in U.S.**

Fig 8 (a) shows that the U.S. domain average TDEP of TIN generally decreased over the past two decades, from 5.55 kg N ha yr$^{-1}$ in 1990 to 5.00 kg N ha yr$^{-1}$ in 2010. The decrease in TIN TDEP is mainly caused by reductions in TNO$_3$. The TNO$_3$ WDEP were estimated to decrease from 1.26 kg N ha yr$^{-1}$ to 0.76 kg N ha yr$^{-1}$, and TNO$_3$ DDEP decreased from 1.98 kg N ha yr$^{-1}$ to 1.35 kg N ha yr$^{-1}$, during the same period. DDEP accounts for large fractions of TDEP for TIN over the entire 1990 to 2010-

time period, 58%-65% of TDEP over the U.S. (supporting Fig. S5). The relative proportions of TNO$_3$ over the TDEP have also changes over the past 2 decades in response to changes in precursor emissions. TNO$_3$ deposition dominates TIN TDEP till the early 2000s. After 2003, however, the NHx dominates the TIN TDEP over the U.S. (supporting Fig. S5). This is consistent with Li et al. (2016) who showed that the U.S. TIN deposition has transitioned from being dominated by TNO$_3$ to NH$_x$ as a result of NOx emission reductions and increases of unregulated NH$_3$ emissions. The increasing contributions of NH$_X$

to the TIN TDEP can also be seen from Fig. 9, which shows increasing proportions of NH$_X$ contributions across larger regions of the continental U.S. during the 1990-2010 period (significant increasing trend ($p < 0.05$) for the NHx fraction of the total TIN across the U.S.). This has resulted from the significant NO$_X$ reduction due to regulations and growth in NH$_3$ emission (Warner et al., 2017).





Similar to TIN TDEP, the $TSO_X$ TDEP has also decreased, from 6.85 kg S $ha^{-1}$ $yr^{-1}$ in 1990 to 3.26 kg S $ha^{-1}$ $yr^{-1}$ in 2010, as a result of the decreasing anthropogenic $SO_2$ emissions (Smith et al., 2011; Xing et al., 2013). The $TSO_X$ DDEP dominates the $TSO_X$ TDEP during the first decade, but $TSO_X$ WDEP becomes dominant after the year 2004. The dry sulfur deposition has decreased by 58% from 1990 to 2010, from 3.65 kg S $ha^{-1}$ $yr^{-1}$ to 1.55 kg S $ha^{-1}$ $yr^{-1}$, while the wet sulfur deposition has

decreased by 47%, from 3.20 kg S $ha^{-1}$ $yr^{-1}$ to 1.70 kg S $ha^{-1}$ $yr^{-1}$ during the same period.

**Conclusion**

In this study, we used model simulations spanning a 21-year period from 1990-2010 to investigate the spatial distribution and temporal trends in the total inorganic nitrogen (TIN) and total sulfur ($TSO_X$) deposition across the U.S., including changes in chemical composition of the deposition as well as relative importance of the wet (WDEP) and dry deposition (DDEP)

components. By evaluating the model's performance against observation from the NADP network, we found that the model generally underestimated the WDEP for both the oxidized nitrogen ($TNO_3$) deposition and reduced nitrogen ($NH_X$) deposition across the U.S. The model underestimated $TSO_X$ WDEP in the eastern U.S., but overestimated it in the western U.S. The model exhibited better performance in simulating the WDEP in the eastern U.S. than in the western U.S. The 21-yr model simulations captured the spatial pattern of decreasing trends for the WDEP of $TNO_3$ and $TSO_X$ very well, with a correlation coefficient

typically larger than 0.9. However, the model generally underestimated the decreasing trends of the $TNO_3$ and $TSO_X$ WDEP. The model performance is worse in simulating the spatial distribution and trends of the $NH_X$ deposition compared with $TNO_3$ and $TSO_X$, which may be caused by uncertainties in the representation of $NH_3$ emissions in the model. The underestimation of the $NH_X$ deposition could also be caused by uncertainties in temporal and spatial representation of emissions associated with fertilizer applications and bi-directional exchange of $NH_3$ between the air and underlying soil and vegetation surfaces.

Applying the bi-directional $NH_3$ exchange mechanism in the coupled model could improve the model's ability in simulating $NH_X$ deposition (Appel et al., 2011; Bash et al., 2013).

The modeled total deposition (TDEP) of TIN and $TSO_4$ is higher in the eastern U.S. and lower in the western U.S. For TIN, it is highest in the Eastern Temperate Forests and lowest in the Northwestern Forested Mountains. For $TSO_X$ it is also highest in the Eastern Temperate Forests but lowest in the North American Deserts. The TDEP of TIN has seen significant decreasing

trends over Eastern Temperate Forests, Northern Forests, Mediterranean California and Marine West Coast Forest, and results from decreases in both wet and dry deposition of $TNO_3$. Modeled TDEP of $TSO_X$ was found to be decreasing over the entire U.S., with larger decreasing trends for the dry deposition compared with the wet deposition.




The TDEP of TIN over the entire U.S. domain was dominated by DDEP, accounting from 58%-65% of the total from 1990 to 2010. TDEP of oxidized nitrogen dominated TIN deposition in the U.S. in the first decade but a shift occurred in 2003 when TDEP of reduced nitrogen became the dominant factor. The DDEP of $TSO_4$ dominates the total sulfur deposition in the first decade while WDEP becomes the dominant factor after the year 2004.

Our analysis as well as others (Li et al., 2016; Kharol et al., 2017) show that reduced nitrogen has dominated the total nitrogen deposition budget in the U.S. in recent years. Additionally, model calculations show strong increasing trends in dry deposition amounts of $NH_x$ across the U.S. which arise both from increasing $NH_3$ emissions but also perhaps from reduced transport distances. Reductions in $SO_2$ and $NO_x$ emissions (and consequently their oxidation products) have decreased the amounts of $NH_x$ partitioning to the aerosol phase where scavenging by rain is the primary sink. Consequently, more $NH_x$ remains in the

gas-phase and dry deposits closer to the source regions. The study highlights the growing importance of $NH_X$ deposition as emissions of $NO_X$ and $SO_2$ have been reduced substantially over the years. We conclude that it is urgent to acquire accurate $NH_3$ emissions inventories and maintain additional measurements of $NH_X$, not only for improving the air quality model's performance, but also for controlling the nitrogen deposition in the U.S. In addition, dry deposition of $TNO_3$ and $TSO_4$ is a large fraction of the total deposition in the U.S., demonstrating the need for accurate dry deposition measurements, as well as

more robust characterization of dry deposition in air quality models.

**Data availability**: The wet deposition data from the U.S. National Atmospheric Deposition Programm can be downloaded from the website (http://nadp.sws.uiuc.edu/). The 21-yrs model outputs for the coupled WRF-CMAQ model can be shared by contacting the corresponding author (Y. Zhang, zhang.yuqiang@epa.gov, yuqiangzhang.thu@gmail.com).

**Competing interests**. The authors declare that they have no conflict of interest.

**Acknowledgement:** This research was supported in part by an appointment to the Research Participation Program at the U.S. EPA, Office of Research and Development (ORD), administered by the Oak Ridge Institute for Science and Education

(ORISE) through an interagency agreement between the U.S. Department of Energy and the U.S. EPA. We greatly acknowledge James Kelly and Kristen Foley from US EPA for their comments and suggestions on the improvements of initial version of this manuscript.

**Disclaimer**: The views expressed in this paper are those of the authors and do not necessarily represent the view or policies



of the U.S. Environmental Protection Agency.

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



## Tables and Figures

Table 1. Correlation coefficient (R), mean bias (MB, kg ha$^{-1}$), and normalized mean bias (NMB, %) for all the annual accumulated wet deposition (WDEP) between the model and NADP sites from 1990 to 2010, including both model values with and without applying monthly/annual precipitation adjustment.

| | | TNO$_3$ | NH$_x$ | TSO$_x$ |
|---|---|---|---|---|
| **R** | No adjustment | 0.89 | 0.77 | 0.92 |
| | Monthly Precip-adjust | 0.91 | 0.81 | 0.94 |
| | Annual Precip-adjust | 0.92 | 0.81 | 0.94 |
| **MB** | No adjustment | -1.92 | -0.50 | -0.37 |
| | Monthly Precip-adjust | -1.89 | -0.52 | -0.53 |
| | Annual Precip-adjust | -2.16 | -0.56 | -0.77 |
| **NMB** | No adjustment | -31.6 | -30.9 | -5.1 |
| | Monthly Precip-adjust | -32.1 | -33.7 | -7.5 |
| | Annual Precip-adjust | -35.6 | -35.1 | -10.5 |
| **R for trends** | No adjustment | 0.85 | 0.35 | 0.86 |
| | Monthly Precip-adjust | 0.94 | 0.64 | 0.95 |
| | Annual Precip-adjust | 0.94 | 0.66 | 0.95 |

Table 2. Evaluation results for 10 ecoregions for TNO$_3$ WDEP. The units for the means and MB are kg ha$^{-1}$, and kg ha$^{-1}$ yr$^{-1}$ for the trends. The bolded values indicate trends that are statistically significant with the P value less than 0.05 for the standard Student's t test.

| ID | Regions Name | #sites | Mean | | MB | NMB | R | Trends | |
|---|---|---|---|---|---|---|---|---|---|
| | | | Obs | Mod | | | | Obs | Mod |
| 5 | Northern Forests | 18 | 7.56 | 4.97 | -2.59 | -0.34 | 0.93 | **-0.22** | **-0.16** |
| 6 | Northwestern Forested Mountains | 28 | 3.23 | 1.28 | -1.95 | -0.60 | 0.70 | -0.03 | **-0.01** |
| 7 | Marine West Coast Forest | 3 | 1.55 | 2.34 | 0.79 | 0.51 | 0.44 | **-0.02** | 0.01 |
| 8 | Eastern Temperate Forests | 72 | 8.77 | 6.14 | -2.63 | -0.30 | 0.97 | **-0.20** | **-0.17** |
| 9 | Great Plains | 24 | 4.73 | 2.62 | -2.11 | -0.45 | 0.87 | **-0.05** | **-0.04** |
| 10 | North American Deserts | 17 | 1.81 | 0.66 | -1.15 | -0.63 | 0.82 | **-0.02** | **-0.01** |
| 11 | Mediterranean California | 4 | 2.34 | 2.39 | 0.05 | 0.02 | 0.76 | **-0.09** | -0.03 |
| 12 | Southern Semi-arid Highlands | 1 | 1.59 | 0.49 | -1.11 | -0.69 | 0.85 | -0.02 | -0.01 |
| 13 | Temperate Sierras | 2 | 2.49 | 0.80 | -1.68 | -0.68 | 0.61 | -0.01 | 0.00 |
| 15 | Tropical Wet Forests | 1 | 5.80 | 2.07 | -3.73 | -0.64 | 0.88 | **0.11** | **0.04** |



Table 3. The same as Table 2 but for NH$_X$.

| ID | Regions Name | #sites | Mean | | MB | NMB | R | Trends | |
| | | | Obs | Mod | | | | Obs | Mod |
|---|---|---|---|---|---|---|---|---|---|
| 5 | Northern Forests | 18 | 1.92 | 1.22 | -0.7 | -0.37 | 0.83 | -0.01 | -0.01 |
| 6 | Northwestern Forested Mountains | 28 | 0.64 | 0.39 | -0.26 | -0.4 | 0.36 | 0.00 | **0.00** |
| 7 | Marine West Coast Forest | 3 | 0.45 | 0.69 | 0.24 | 0.54 | 0.16 | 0.00 | **0.01** |
| 8 | Eastern Temperate Forests | 72 | 2.13 | 1.58 | -0.55 | -0.26 | 0.66 | 0.00 | 0.00 |
| 9 | Great Plains | 24 | 2.03 | 0.91 | -1.12 | -0.55 | 0.86 | **0.03** | **0.01** |
| 10 | North American Deserts | 17 | 0.58 | 0.19 | -0.38 | -0.66 | 0.62 | 0.00 | 0.00 |
| 11 | Mediterranean California | 4 | 1.01 | 0.64 | -0.38 | -0.37 | 0.82 | -0.02 | 0.00 |
| 12 | Southern Semi-arid Highlands | 1 | 0.42 | 0.13 | -0.29 | -0.69 | 0.76 | 0.00 | 0.00 |
| 13 | Temperate Sierras | 2 | 0.63 | 0.26 | -0.37 | -0.59 | 0.75 | 0.00 | 0.00 |
| 15 | Tropical Wet Forests | 1 | 1.14 | 0.33 | -0.81 | -0.71 | 0.75 | **0.04** | **0.01** |

Table 4. The same as Table 2 but for TSO$_X$.

| ID | Regions Name | #sites | Mean | | MB | NMB | R | Trends | |
| | | | Obs | Mod | | | | Obs | Mod |
|---|---|---|---|---|---|---|---|---|---|
| 5 | Northern Forests | 18 | 7.76 | 7.33 | -0.42 | -0.06 | 0.95 | **-0.29** | **-0.23** |
| 6 | Northwestern Forested Mountains | 28 | 2.15 | 1.88 | -0.27 | -0.13 | 0.70 | **-0.05** | **-0.01** |
| 7 | Marine West Coast Forest | 3 | 3.35 | 6.08 | 2.73 | 0.82 | 0.46 | -0.02 | 0.04 |
| 8 | Eastern Temperate Forests | 72 | 11.78 | 11.04 | -0.70 | -0.06 | 0.97 | **-0.34** | **-0.29** |
| 9 | Great Plains | 24 | 4.16 | 2.95 | -1.21 | -0.29 | 0.91 | **-0.07** | **-0.04** |
| 10 | North American Deserts | 17 | 1.38 | 0.81 | -0.58 | -0.41 | 0.79 | **-0.04** | -0.01 |
| 11 | Mediterranean California | 4 | 1.40 | 3.15 | 1.75 | 1.25 | 0.67 | **-0.03** | 0.01 |
| 12 | Southern Semi-arid Highlands | 1 | 1.45 | 0.89 | -0.56 | -0.39 | 0.91 | **-0.07** | **-0.04** |
| 13 | Temperate Sierras | 2 | 2.30 | 1.05 | -1.25 | -0.54 | 0.76 | **-0.08** | -0.01 |
| 15 | Tropical Wet Forests | 1 | 7.41 | 2.94 | -4.47 | -0.60 | 0.73 | 0.09 | 0.04 |



Table 5. TDEP (WDEP+DDEP, units of kg N ha$^{-1}$ for nitrogen deposition including TNO$_3$, NH$_X$ and TIN, and kg S ha$^{-1}$ for TSO$_4$) in 1990 and 2010 for the 10 ecoregions.

| ID | Regions Name | TNO$_3$ | | NH$_X$ | | TIN | | TSO$_4$ | |
|----|--------------|---------|---------|--------|--------|------|------|---------|---------|
| | | 1990 | 2010 | 1990 | 2010 | 1990 | 2010 | 1990 | 2010 |
| 5 | Northern Forests | 4.21 | 2.19 | 2.35 | 2.56 | 6.56 | 4.74 | 9.86 | 3.56 |
| 6 | Northwestern Forested Mountains | 1.36 | 1.12 | 0.91 | 1.26 | 2.27 | 2.38 | 1.75 | 1.47 |
| 7 | Marine West Coast Forest | 1.07 | 1.35 | 2.00 | 2.43 | 3.7 | 3.78 | 5.03 | 3.95 |
| 8 | Eastern Temperate Forests | 6.12 | 3.27 | 3.96 | 4.68 | 10.08 | 7.94 | 17.54 | 6.66 |
| 9 | Great Plains | 2.45 | 1.84 | 2.77 | 3.97 | 5.22 | 5.81 | 3.36 | 2.16 |
| 10 | North American Deserts | 1.49 | 1.12 | 0.83 | 1.01 | 2.32 | 2.13 | 1.34 | 1.05 |
| 11 | Mediterranean California | 3.15 | 2.08 | 2.68 | 3.36 | 5.84 | 5.44 | 1.68 | 1.74 |
| 12 | Southern Semi-arid Highlands | 1.68 | 1.10 | 1.18 | 0.93 | 2.86 | 2.03 | 2.87 | 0.92 |
| 13 | Temperate Sierras | 2.00 | 1.48 | 0.91 | 1.02 | 2.91 | 2.5 | 2.33 | 1.2 |
| 15 | Tropical Wet Forests | 4.11 | 3.35 | 1.27 | 2.05 | 5.38 | 5.41 | 5.15 | 3.77 |

Table 6. Trends for total deposition (WDEP+DDEP, units of kg N ha$^{-1}$ yr$^{-1}$ for nitrogen deposition including TNO$_3$, NH$_X$ and TIN, and kg S ha$^{-1}$ yr$^{-1}$ for TSO$_4$) over the ten ecoregions. The bolded values indicate trends that are statistically significant
10 with the P value less than 0.05 for the Student's t test.

| ID | Regions Name | TNO$_3$ | NH$_X$ | TIN | TSO$_4$ |
|----|--------------|---------|--------|-----|---------|
| 5 | Northern Forests | **-0.087** | **0.016** | **-0.071** | **-0.23** |
| 6 | Northwestern Forested Mountains | **-0.013** | **0.011** | -0.002 | **-0.021** |
| 7 | Marine West Coast Forest | **-0.018** | 0.002 | **-0.017** | **-0.053** |
| 8 | Eastern Temperate Forests | **-0.15** | **0.034** | **-0.12** | **-0.51** |
| 9 | Great Plains | **-0.041** | **0.044** | 0.003 | **-0.082** |
| 10 | North American Deserts | **-0.016** | **0.008** | -0.008 | **-0.023** |
| 11 | Mediterranean California | **-0.051** | 0.013 | **-0.038** | -0.013 |
| 12 | Southern Semi-arid Highlands | **-0.014** | 0.002 | -0.012 | **-0.074** |
| 13 | Temperate Sierras | **-0.016** | **0.009** | -0.006 | **-0.054** |
| 15 | Tropical Wet Forests | **-0.026** | **0.041** | 0.015 | **-0.055** |





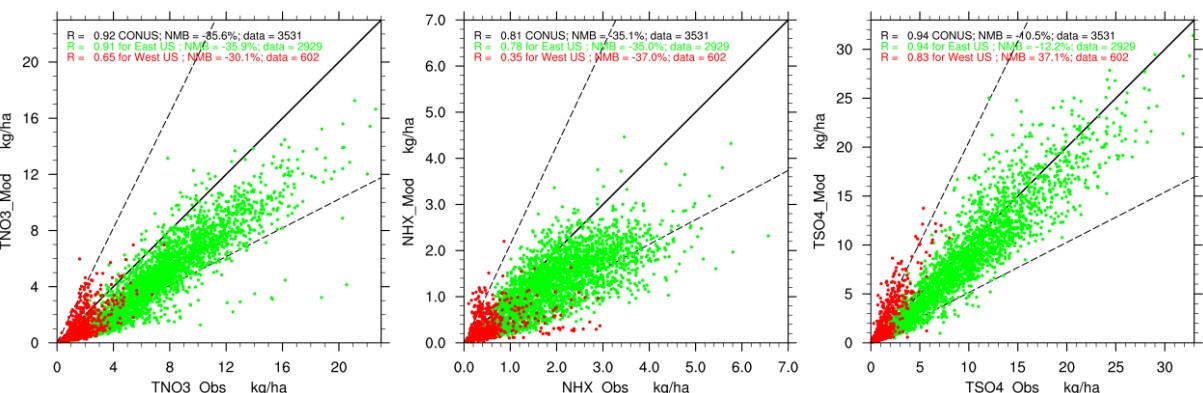

Figure 1. Scatter plots for the annual accumulated WDEP (total oxidized nitrogen ($TNO_3$), reduced nitrogen ($NH_X$), and total sulfate ($TSO_X$)) between observations and precipitation-adjusted model results from 1990 to 2010 for 170 valid sites with 3531 valid data points. Each NADP is assumed to be valid for our analysis only if at least 18 years of observation data are available

5    at that site and the data coverage is at least 75% for each year. Each point in the plots represents the annual accumulated WDEP for a given site and year. Note that the annual accumulated WDEP values used in this analysis may not be the actual annual totals due to missing data in the observations.




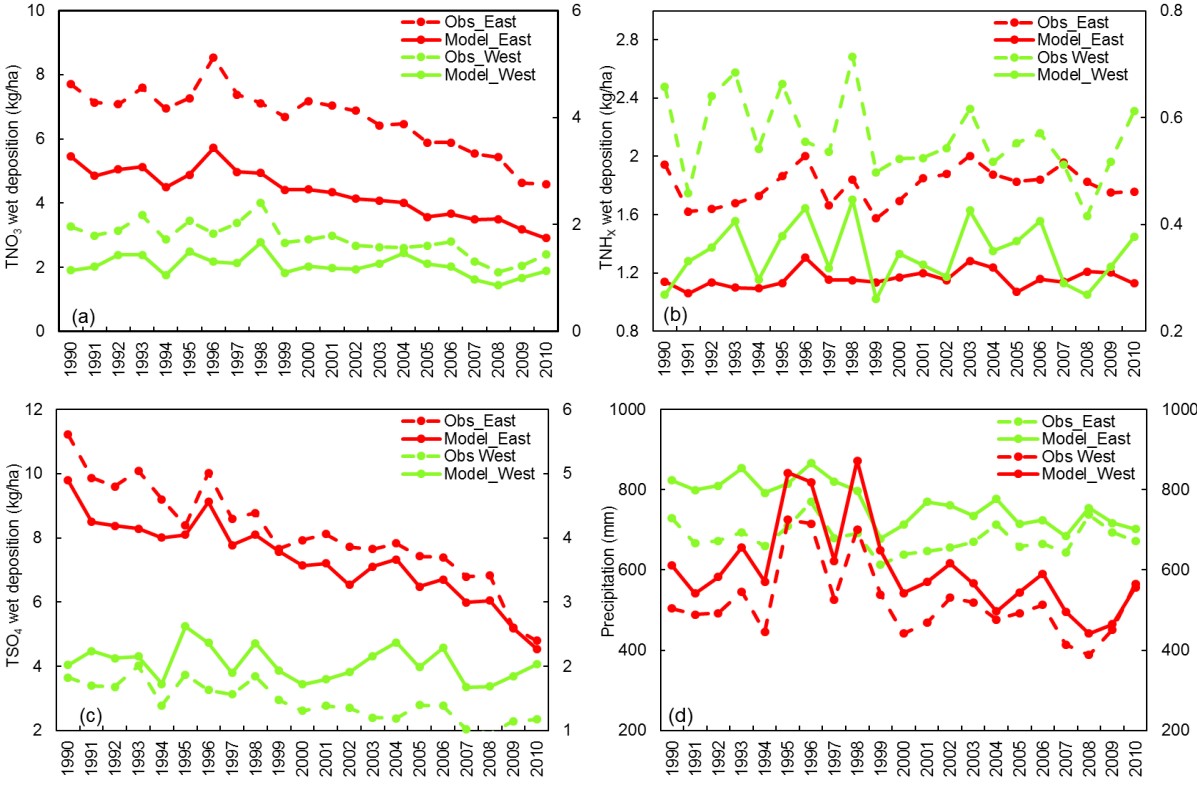

Figure 2. Comparison of the temporal trends for the annual accumulated WDEP (across all the 170 valid sites) of (a) TNO$_3$,
(b) TNH$_X$, (c) TSO$_4$, and (d) annual accumulated precipitation, for the eastern US (green, averaged over 141 sites) and western
US (red, average over 29 sites) between observation (dashed lines) and annual precipitation-adjusted model values (solid lines).
The scale shown on the left is for the eastern US, and on the right for the western US.

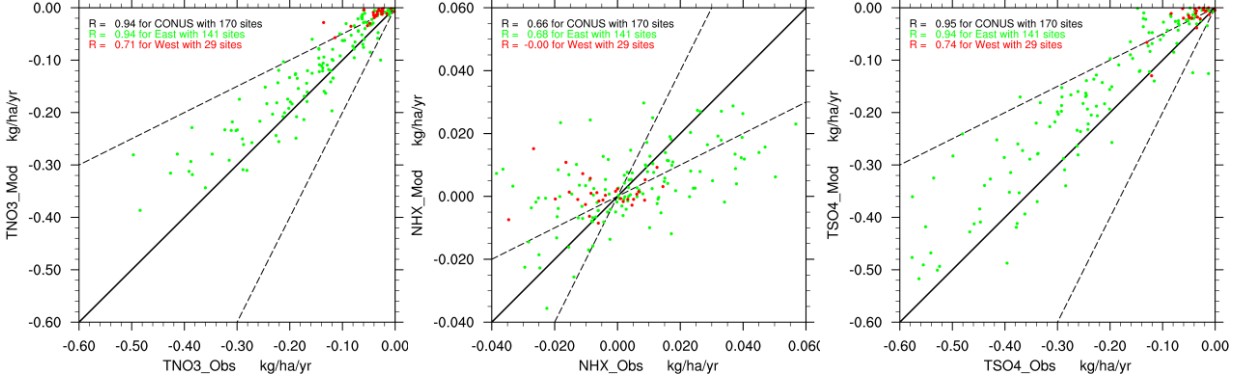





Figure 3. Comparison of the WDEP trend for each valid site between observations and precipitation-adjusted model values. Each NADP site is assumed to be valid for our analysis only if at least 18 years of observation data are available at that site and the data coverage is at least 75% for each year.

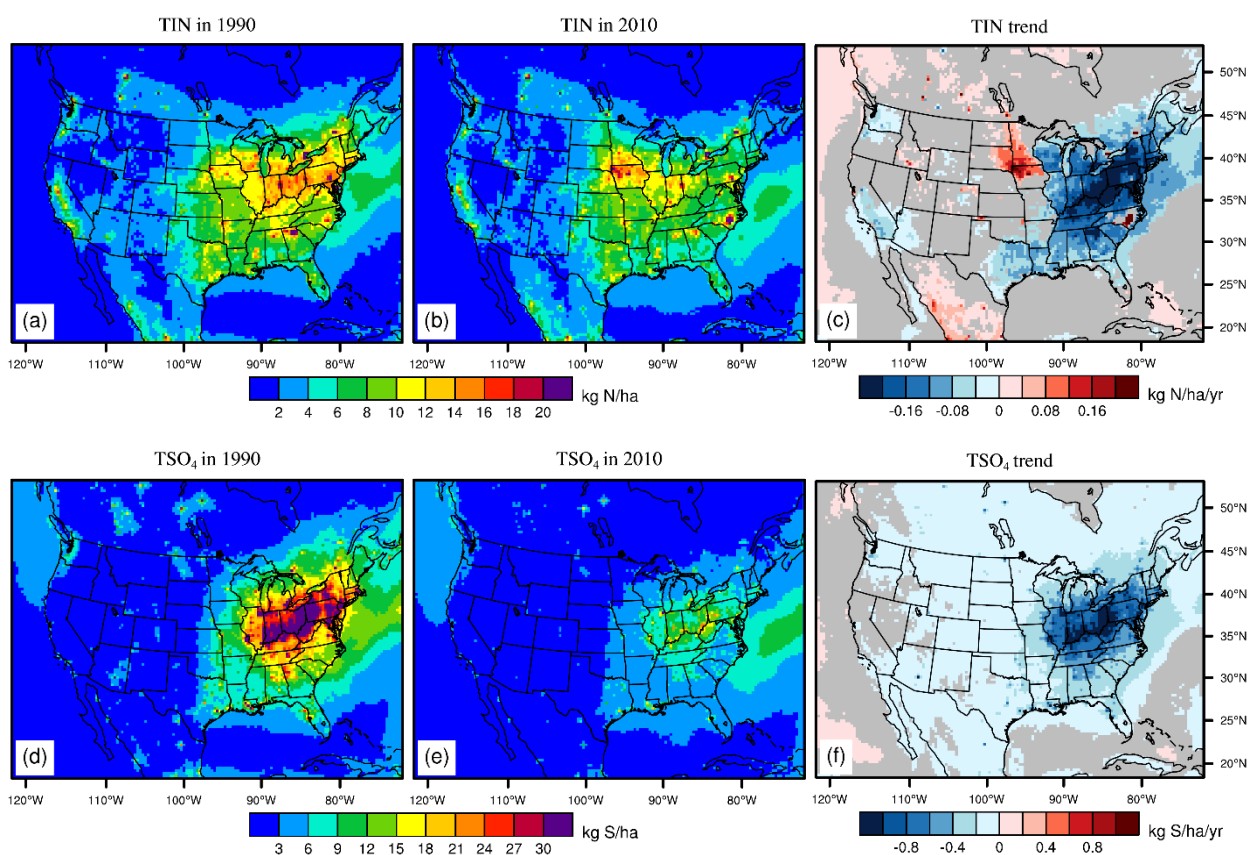

Figure 4. Spatial distribution of annual TDEP of total inorganic nitrogen (TIN, kg N/ha, top panel) and sulfur (kg S/ha, bottom panel) in 1990 (a, d), 2010 (b, e), and the simulated trends of the TIN (c, kg N ha$^{-1}$ yr$^{-1}$) and total sulfur (f, kg S ha$^{-1}$ yr$^{-1}$) TDEP changes over the 2 decades. Grey areas on the right plot show p value for the standard two-tailed Student T-test greater than 0.05 (i.e. areas where trend estimates were not significant at the 95% confidence level).



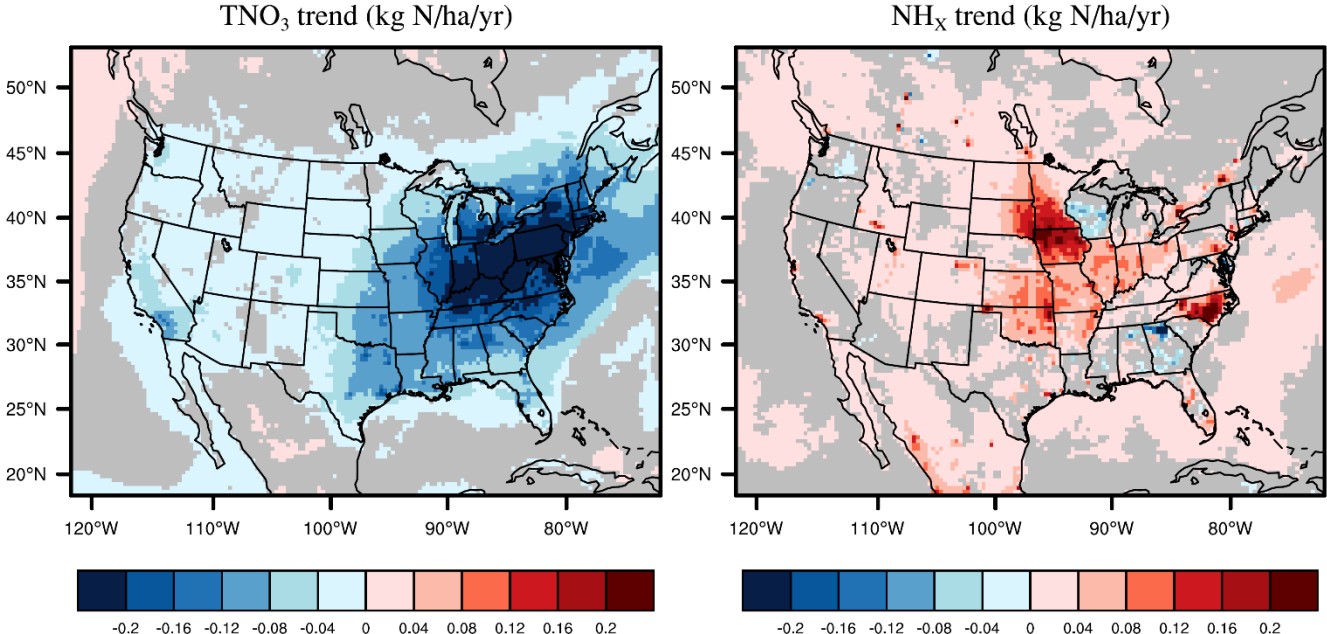

Figure 5. Spatial distribution of the trends for the TDEP of total oxidized nitrogen deposition (TNO$_3$ on the left), and reduced nitrogen (NH$_X$ on the right) from 1990 to 2010. Grey areas on the right plot show p value great than 0.05 for the standard two-tailed Student T-test (i.e. areas where trend estimates were not significant at the 95% confidence level).



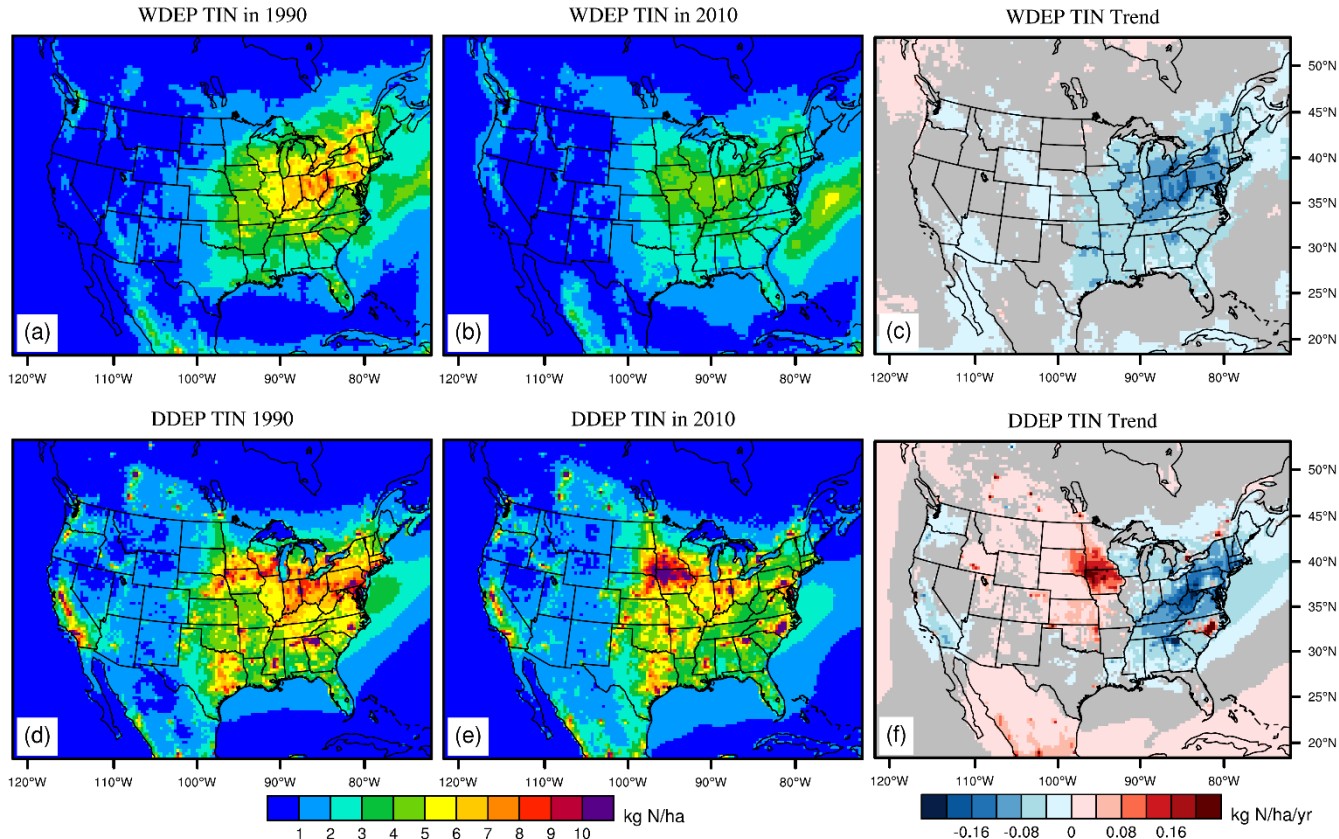

Figure 6. Spatial distribution of WDEP (top panel) and DEP (bottom panel) of TIN (kg N ha$^{-1}$) in 1990 (a, d), 2010 (b, e), and the simulated trends (c, f, kg N ha$^{-1}$ yr$^{-1}$) over the 2 decades. Grey areas on the right plot show p value great than 0.05 for the standard two-tailed Student T-test (i.e. areas where trend estimates were not significant at the 95% confidence level).





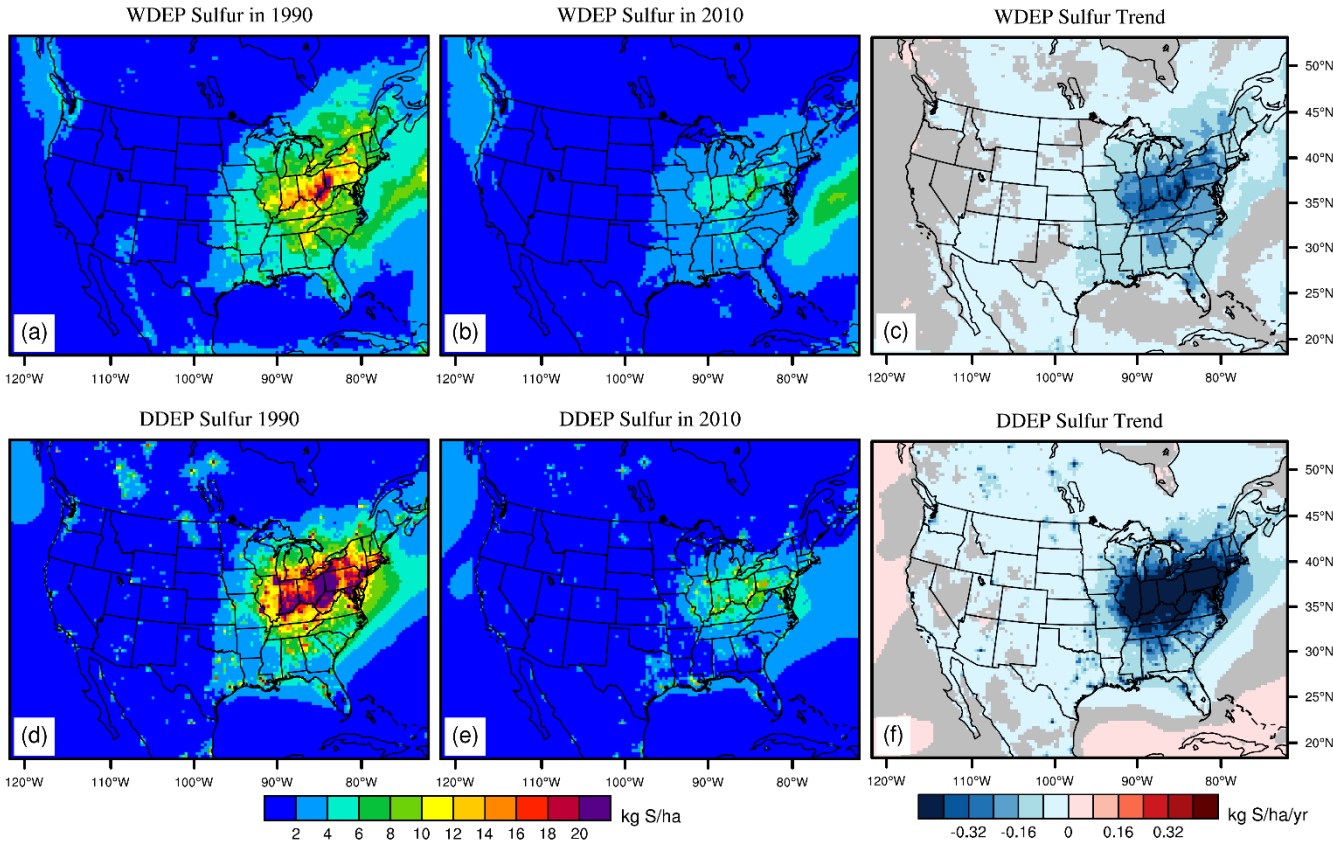

Figure 7. As in Fig. 6 but for sulfur. The units are kg S ha$^{-1}$ for (a, b, d, e) and kg S ha$^{-1}$ yr$^{-1}$ for (c, f).





Figure 8. Interannual variability of the TDEP for inorganic nitrogen (a), and sulfur (b) in the US from 1990 to 2010, including their fractions (WDEP of oxidized nitrogen, WDEP of reduced nitrogen, DDEP of oxidized nitrogen and DDEP of reduced nitrogen deposition for the nitrogen, and WDEP versus DDEP for sulfur).



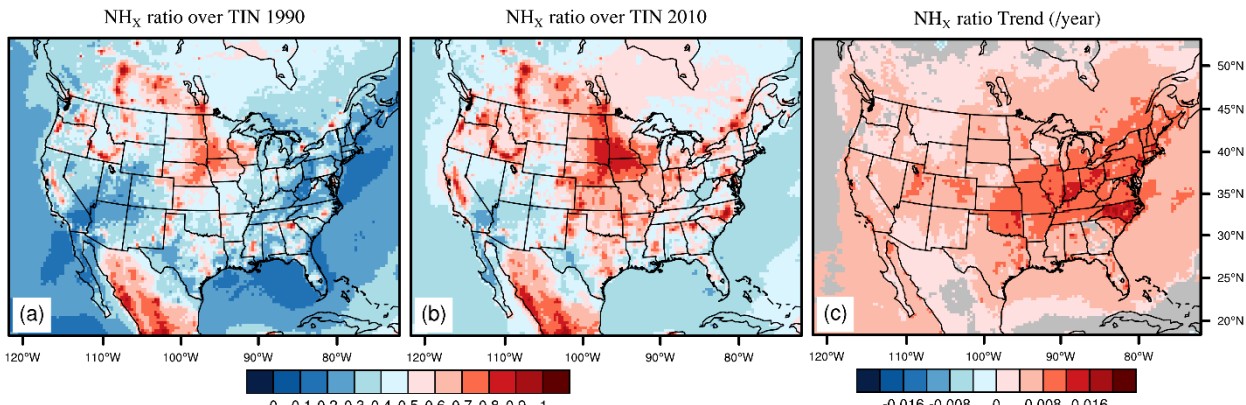

Figure 9. The ratio of TDEP of $NH_X$ over the TDEP of TIN in 1990 (a), 2010 (b), and the trend (c). The blue color in (a,b) indicates an NHx ratio less than 0.5 which means $TNO_3$ dominates the total nitrogen deposition, while the red color indicates a ratio larger than 0.5, and $NH_X$ dominates the total nitrogen deposition.