# Peer review of "Long-term trends in total inorganic nitrogen and sulfur deposition in the U.S. from 1990 to 2010"

_Atmospheric Chemistry and Physics, 2018_

## Referee Comment (RC1) · Anonymous Referee #1 · 19 Mar 2018

The manuscript discussed twenty year trends in nitrogen (N) and sulfur (S) deposition in the U.S. based on the WRF-CMAQ model simulations. The article identified the current limitations of modeling nitrogen and sulfur deposition and discussed spatial distributions and trends of those species in the U.S. Those results confirmed that reduced nitrogen had dominated the total nitrogen deposition in the U.S. and highlighted the necessity of controlling reduced N. The structure of the manuscript, the results and the presentation of the material are reasonably good. The topic is relevant and certainly deserves publication in Atmospheric Chemistry and Physics. There are, however, several changes and additions required before publication. Specific comments: Page 2, Line 5-6: Please split up these references so that they are associated with the specific impacts being discussed, rather than all placed at the end of the sentence. And, I do

not think increased sulfur deposition could cause aquatic eutrophication. Page 2, Line 20: Change "pattern" to "patterns". Page 2, Line 26: Please explain "complexity" more. Page 3, Line 20: Change "description" to "descriptions". Page 3, Line 22: Add "supporting" in the front of "Table S1". Page 3, Line 22: O3 and PM2.5 should be defined at first mention. Page 3, Line 24-25: Provide some references. Page 3, in section 2.1: The authors should specify how the dry depositions were estimated. Page 4, Line 11: I am not sure whether the 110th meridian west is appropriate to divide east and west. There are more sites in the east than the west if 110oW is used. A map with 110oW and all the sites should be included in the supplement. Page4, Line 13: How did the authors get the value of 0.984? Page 5, In 3.1, Could the authors be more specific about the improvements since Appel et al (2011)? Page 5, It seems like the authors only did model evaluation and model justification for wet deposition. How was model performance for dry deposition? The authors could use data from AMON, IMPOROVE and EPA CASTNET to do this work. Page 5, Line 19: Change "models" to "model results" Page 6, Line 15: It should be "Table S3" Page 6, Line 26 – Page 7, Line 2: Please explain the reasons for those results. Page 7, Line 11- 20: Which one dominates the decrease of TSOx, SO42- or SO2 ? Page 26, The legend of Fig 8 (a) needs to be fixed.

---

## Referee Comment (RC2) · Anonymous Referee #2 · 8 Apr 2018

This paper examines trends in inorganic nitrogen and sulfur deposition from 1999 to 2010 across the U.S. This analysis is performed using WRF-CMAQ model simulations. The results from the model are compared to data from the NADP (National Atmospheric Deposition Program) Network. The trends and spatial patterns observed are discussed.

Overall, this is a good paper. But I do have some concerns. I feel a large part of the methods section is missing as the authors do not actually discuss the dry deposition data being used. The authors establish how wet deposition from the model compares with the observations, but they do not actually discuss what might lead to the differences in the two until a few lines in the conclusions. More importantly, they do not make this same comparison for the dry deposition results, yet they go on to discuss

the trends in dry deposition from the model in the paper. All of this is outlined in detail below in my comments, which need to be addressed before this paper can be considered for publication

General Comments: -I am a bit surprised that the abbreviation TSOx is used for sulfur deposition rather than TS. TS to me seems more fitting, but I understand if the other is more traditionally used as I am not as familiar with that literature as I am with nitrogen deposition. However, that being said it seems that the paper goes back and forth using TSOx and TSO4 to represent sulfur deposition. This is true throughout the main text, figures, and supporting information. This should be checked.

Specific Comments: Abstract Page 1, Line 12 – The abbreviation WRF-CMAQ is not defined

Page 1, Lines 15-17 – The authors mention that the model generally underestimates the wet deposition. But they do not provide any reasons why this is. This should be added to the abstract.

Page 1, Line 19 – Suggest changing decrease of TNO3 to decreases in TNO3

Page 1, Line 20 – The authors mention there are increasing trends in TIN deposition over the Tropical Wet Forest. This is the only region type listed in the text that does not have a geographic location included in its title. I think this makes it hard for readers to understand where it is. I would suggest adding a phrase such as southern Florida to aid the reader.

Page 1, Line 22 – Suggest removing the words region of before Eastern

Page 1, Line 23 – Suggest removing the words region of before Tropical

Page 1, Line 28 – Suggest adding an a before combination

1.Introduction Page 2, Line 12 – Suggest changing twice higher than to twice as high as

Page 2, Line 13 – Suggest removing the the before sulfur

Page 2, Line 14 – form fossil-fuel should be from fossil-fuel

Page 2, Lines 15-20 – Here the authors discuss the wet deposition national networks. But they do not actually tell how the measurements are made. I would suggest adding some text telling how the samples are collected and then measured by ion chromatography to provide the data.

Page 2, Line 21 – Suggest adding the words e.g., before EEA. Also a comma is missing after EEA

Page 2, Line 22 – Suggest removing the comma and phrase to name a few after 2015

Page 3, Line 2 – There is an extra period after loss

Page 3, Line 5 – Suggest removing the second Zhao et al.

Page 3, Line 8 – Suggest changing process to processes

2.Methods 2.1.Model setup Page 3, Line 16 – The abbreviation WRF-CMAQ is not defined

Page 3, Line 20 – There is an extra comma after Gan et al. Also suggest adding an a before detailed

Page 3, Line 22 – The chemical abbreviation used are not defined

Page 3, Line 24 – Suggest changing was shown to has shown

2.2.Wet deposition observations in the U.S. Page 4, Line 11 – Suggest changing observation data used for to observational data used for the

Page 4, Line 13 – Suggest changing combine WDEP to combines WDEP and with 0.984 to by 0.984

Page 4, Line 15 – Suggest changing combine WDEP to combines WDEP and with 1.06

to by 1.06

Page 4, Line 17 - Suggest changing combine WDEP to combines WDEP and with 1.50 to by 1.50

Page 4, Lines 2-26 – Why is there no section on dry deposition in the Methods section? The authors explicitly state in the introduction that there are no direct measurements of this, but that they are calculated at some sites. So then information on how they are calculated and what is used here should be provided to the reader so that they fully understand the analysis that is being performed.

Page 4, Lines 2-26 – The authors do not actually explain how the data from the network is collected. I understand the model analysis is the point of the paper. But since these observational data are used to evaluate the model then the authors should provide at least some text to give the readers context.

Page 5, Line 5 – There is an extra comma after equation 2

Page 3, Line 14 to Page 5, Line 6 – In the methods section there is no discussion of the trend analysis that is used throughout the paper. What is this analysis? How is it done? This should be added to the paper.

3.Results 3.1.Model evaluation of WDEP Page 5, Line 16 – Suggest changing increases for all the three to increase for all three. Also exhibit should be exhibited. Also suggest changing in east than that in west to in the east than the west

Page 5, Line 17 – A period is missing after (Appel et al., 2011)

Page 5, Line 19 – Suggest changing both observations and models to both the observations and model results

Page 5, Line 20 – Suggest adding a the before Tropical

Page 6, Line 11 – I am not sure I understand the phrase but a slightly distinctions in trends for different ecoregions. Is it maybe but with slight distinctions in the trends for

each ecoregion?

Page 6, Lines 11-13 – The authors mentions that the model generally underestimates decreasing WDEP trends for all sites, but for NHx they see increasing WDEP trends. Why is this? The authors need to tell why they think this might be the case for the model.

Page 6, Line 14 – Suggest removing the word results before model

Page 6, Line 15 – Suggest changing increases for all the three to increase for all three. Also why are the authors only looking at the data from 2002-2006 when they discuss the NMB increase observed? This needs to be clarified.

Page 6, Line 18 – Suggest changing are more to have more

Page 6, Line19 – Suggest changing challenging to challenges

Page 5, Line 8 to Page 6, Line 20 – Why is there no matching section on the model evaluation for DDEP? The remainder of the results section discusses the trends in total, wet, and dry deposition so it seems that it should be established how the model compares with the calculated dry deposition values provided by NADP.

3.2.Spatial patterns of modelled total deposition of nitrogen and sulfur Page 6, Line 21 – modelled should be written as modeled to be consistent with how it is used throughout the rest of the text

Page 7, Line 18 – Suggest removing the and after showed

Page 7, Line 19 – Believe that Table 4 should be Table 6

3.3.Wet versus dry nitrogen and sulfur deposition trends in the U.S. Page 7, Line 25 – Suggest adding a the before Eastern

Page 7, Line 26 – Suggest adding a the before Northern and Great

Page 7, Line 27 – Suggest changing was mainly to were mainly

Page 7, Line 28 – The authors mention that there are no significant changes for WDEP of NHx. However, in Table S4 the values for Tropical Wet Forests are in bold, which is what indicates a significant trend. Also there is light blue being shown in Figure S4b. This needs to be clarified.

Page 8, Line 8 – Suggest adding an a before distinct and changing value to values

Page 8, Line 9 – Suggest adding a the before vicinity and changing source to sources

3.4.Deposition budget in U.S. Page 8, Line 18 – Suggest changing were estimated to was estimated

Page 8, Line 19 – Suggest removing the hyphen after 2010

Page 8, Line 21 – Suggest changing changes to changed

Page 8, Line 22 – Suggest changing till to until and removing the the before NHx

Page 8, Line 26 – Suggest changing 1999-2010 to 1999 to 2010

Page 8, Line 27 – Suggest changing emission to emissions

Page 8, Line 28 – The reference is written in blue

Page 9, Line 2 - The references are written in blue

Page 9, Lines 1-5 – I believe that this section is in reference to Figure 8, but there is citation to Figure 8 listed here.

Conclusions Page 9, Line 10 – Suggest changing observation to observations

Page 9, Line 25 – Suggest adding a the before Eastern

Page 10, Line 9 – It should be aerosol-phase

Data availability Page 10, Line 18 – Suggest changing shared to obtained

Competing interests Page 10, Line 21 – Suggest changing conflict to conflicts

Page 10, Line 26 – Suggest adding a the before U.S. and removing the phrase improvements of after suggestions on the

Disclaimer Page 10, Line 28 – Suggest changing view to views

References Page 11, Line 26 – Believe the accent marks in Muller should be over the u

Tables and Figures Table 1 -In first line of caption – Suggest changing for all the annual to for the sum of the annual -In second line of caption - Suggest adding a the before model -What is the difference between R and R for trends? There is no discussion about this in the main text so it is hard to understand why the two set of values are being shown.

Table 2 -In first line of caption – Suggest adding a the before 10 -In third line of caption – There should be a hyphen in t-test -Second column heading – Suggest changing Regions to Region -Third column heading – Suggest changing # sites to # of sites

Table 3 -Second column heading – Suggest changing Regions to Region -Third column heading – Suggest changing # sites to # of sites

Table 4 -Second column heading – Suggest changing Regions to Region -Third column heading – Suggest changing # sites to # of sites

Table 5 -Second column heading – Suggest changing Regions to Region

Table 6 -In third line of caption – There should be a hyphen in t-test -Second column heading – Suggest changing Regions to Region

Figure 1 -In second line of caption – To match the figure between observations and precipitation-adjusted model results should be switched -In third line of caption – Suggest changing Each NADP to The data at each NADP site -Letters should be added to each plot and the caption updated to indicate this -Suggest making a symbol indicating that green is for East sites and red is for West sites as currently this is only

indicated from the small text at the top of each plot -It should be indicated in the caption what the solid and dashed lines in each plot represent -There are no subscripts in the abbreviations used on both the x and y-axes for all plots

Figure 2 -In first line of caption – Suggest changing of (a) TNO3 to for (a) TNO3 -In second line of caption – Suggest removing the phrase annual accumulated before precipitation. -In second, third, and fourth lines of caption - US should be U.S. -There are no x-axis labels -The legend for plots a, b, and c are incorrect as they indicate the data for the East is red and West is green

Figure 3 -In first line of caption – Suggest changing adding a the before observations -In first line of caption – To match the figure between observations and precipitation-adjusted model valves should be switched -In second line of caption – Suggest changing observation to observational -Letters should be added to each plot and the caption updated to indicate this -It should be indicated in the caption what the solid and dashed lines in each plot represent -There are also no subscripts in the abbreviations used on both the x and y-axes for all plots

Figure 4 -In first and second lines of caption – Suggest changing panel to panels -In third line of caption – Suggest changing plot show p value to plots show p values -In fourth line of caption – Suggest adding a comma after i.e. -There are no x and y-axes labels

Figure 5 -In second line of caption – Suggest changing the right plot show p value great than to both plots show p values greater than -In third line of caption – T-test should be t-test. Also suggest adding a comma after i.e. -Letters should be added to each plot and the caption updated to indicate this -There are no x and y-axes labels

Figure 6 -In first line of caption – Suggest changing (top panel) and DEP (bottom panel) to (top panels) and DDEP (bottom panels) -In second line of caption – Suggest changing plot show p value great than to plots show p values greater than -In third line of caption – T-test should be t-test. Also suggest adding a comma after i.e. -There are no

x and y-axes labels

Figure 7 -There are no x and y-axes labels

Figure 8 -In caption – It should be mentioned in the caption that the percent contribution is being indicated on each bar -In first line of caption – US should be U.S. -On the y-axis for both plots, US should be U.S. -Suggest in legend for plot a calling Oxid as NO3 instead and Red as NHx instead so that it matches the main text

Figure 9 -In second line of caption – Suggest changing an NHx to a NHx -In third line of caption – Suggest removing the comma after 0.5 -There are no x and y-axes labels -Title for plot a – Suggest changing NHx ratio over TIN 1990 to TDEP NHx to TIN ratio 1990 -Title for plot b – Suggest changing NHx ratio over TIN 2010 to TDEP NHx to TIN ratio 2010 -Title for plot c – I am not sure I understand this plot title. What is (/year) indicating? Should the title maybe be TDEP NHx to TIN ratio Overall Trend?

Supporting Information Figure S1 -In caption – Suggest changing all the ofs to equal signs (e.g., 5 of Northern Forests to 5 = Northern Forests) -There are no x and y-axes labels -In plot title – US should be U.S. Also what does mask mean? It is not indicated in the caption or text.

Figure S2 -In first line of caption – Suggest changing plot to plots. Also the words observation and model should be switched to match what is actually plotted. -In second line of caption – Suggest changing data. The site in NADP is assumed to data points. The data at each NADP site is assumed -In third line of caption – Suggest changing valid if only at to valid only if at, changing is available to are available, and changing for the to for that -In fourth line of caption – suggest changing plot to plots -In fifth line of caption – Suggest removing the the before missing -In sixth line of caption – Suggest changing observation to observations -It should be indicated in the caption what the solid lines in the plot represent. Also should this be like the other plots and have two dashed lines and one solid line?

Figure S3 -Letters should be added to each plot and the caption updated to indicate this -Suggest making a symbol indicating that green is for East sites and red is for West sites as currently this is only indicated from the small text at the top of each plot -It should be indicated in the caption what the solid and dashed lines in each plot represent -There are no subscripts in the abbreviations used on both the x and y-axes for all plots

Figure S4 -There are no x and y-axes labels -There are no subscriptions in the abbreviations used in the titles for all plots

Figure S5 -In first line of caption – Suggest adding a the before US. Also US should be U.S. -There are no x-axis labels -Suggest changing y-axis labels to Fraction of the Total -Suggest pointing out on both plots somehow 2003 since this is an important year in terms of trends and so that it corresponds with the discussion in the main text. Maybe add a vertical dashed line.

Table S1 -Either the comma or the parenthesis should be removed from Xing et al. reference. Both are not needed.

Table S3 -In third line of caption – Suggest removing the and with at the end of the sentence

Table S4 -In fourth line of caption - There should be a hyphen in t-test -Second column heading – Suggest changing Regions to Region

Table S5 -Second column heading – Suggest changing Regions to Region

[Figure]

---

## Author Comment (AC1) · 29 May 2018

Response to review #2 on acp-2018-116

Long-term trends in total inorganic nitrogen and sulfur deposition in the U.S. from 1990 to 2010

Yuqiang Zhang, Rohit Mathur, Jesse O. Bash, Christian Hogrefe, Jia Xing, Shawn J. Roselle

We thank referee #1 for the positive comments and constructive suggestions, which have helped us improve the manuscript. All referee comments (in blue below) have been carefully addressed, and changes incorporated in the revised manuscript are shown using the track-changes option.

The manuscript discussed twenty year trends in nitrogen (N) and sulfur (S) deposition in the U.S. based on the WRF-CMAQ model simulations. The article identified the current limitations of modeling nitrogen and sulfur deposition and discussed spatial distributions and trends of those species in the U.S. Those results confirmed that reduced nitrogen had dominated the total nitrogen deposition in the U.S. and highlighted the necessity of controlling reduced N. The structure of the manuscript, the results and the presentation of the material are reasonably good. The topic is relevant and certainly deserves publication in Atmospheric Chemistry and Physics.

There are, however, several changes and additions required before publication.
Specific comments:
Page 2, Line 5-6: Please split up these references so that they are associated with the specific impacts being discussed, rather than all placed at the end of the sentence. And, I do not think increased sulfur deposition could cause aquatic eutrophication.
**Response:** We thank the reviewer for pointing this out. We now split the references. The revised sentence is:
"Increased nitrogen and sulfur deposition is detrimental to ecosystems, since it leads to decreased biological diversity (Clark and Tilman, 2008; Clark et al., 2013; Stevens et al., 2004), increased terrestrial and aquatic eutrophication (Bouwman et al., 2002; Bowman et al., 2008; Fisher et al., 2011) and acidification (Greaver et al., 2012; Savva and Berninger, 2010)."

Page 2, Line 20: Change "pattern" to "patterns".
**Response:** We made the change following the reviewer' comment.

Page 2, Line 26: Please explain "complexity" more.
**Response:** We thank the reviewer for pointing this out. Complexity was intended to convey the challenges in spatial interpolation of dry deposition estimates both due to limited availability of observations as well as representativeness of the interpolated fields. To clarify, we have rewritten the sentence as:
"however cannot be easily spatially interpolated due to limited availability of sufficient number of sites in a region as well as the representativeness of the derived fields due to assumptions in the spatial interpolation method (Schwede and Lear, 2014)"

Page 3, Line 20: Change "description" to "descriptions".
**Response:** Thank you for catching the typo. We have revised the sentence as:

"Interested readers are referred to Gan et al. (2015, 2016) for a detailed description of the settings of the CMAQ model"

Page 3, Line 22: Add "supporting" in the front of "Table S1".
**Response:** We thank the reviewer for pointing this out. We now add "supporting" in the front of "Table S1".

Page 3, Line 22: $O_3$ and $PM_{2.5}$ should be defined at first mention.
**Response:** We thank the reviewer for pointing this out. We now added the definition of $O_3$ and $PM_{2.5}$ in the paper:
"The performance of the coupled WRF-CMAQ model for major trace gases, aerosol species and meteorological variables such as ozone ($O_3$), fine particular matter ($PM_{2.5}$)"

Page 3, Line 24-25: Provide some references.
**Response:** We now rewrite the sentence to add the right references to them:
"at both the hemispheric and regional scale has been extensively evaluated in previous studies, and has shown skill in simulating the magnitudes and long-term trends of these variables (Xing et al., 2015a, b; Mathur et al., 2017; Gan et al., 2015, 2016; Astitha et al., 2017)."

Page 3, in section 2.1: The authors should specify how the dry depositions were estimated.
**Response:** We thank the reviewer for the suggestion. The dry deposition for each species is calculated by multiplying the concentration in the lowest model layer by the dry deposition velocity ($V_d$). The dry deposition velocity is calculated as the reciprocal of the sum of the atmospheric ($R_a$, the resistance to transport through the atmosphere above the surface receptors), quasi-laminar boundary layer ($R_b$, the resistance to transport across the thin layer of air that is in contact with the surface and varies with the diffusion of the pollutant transported), and surface resistances ($R_s$, the resistance to the uptake of the pollutant by the surface receptor, typically vegetation or soil). We now add this information on Page 3 line 25:

"The dry deposition of each species in the CMAQ model is calculated by multiplying the concentration in the lowest model layer by the dry deposition velocity ($V_d$). The $V_d$ is calculated as the reciprocal of the sum of the atmospheric ($R_a$, the resistance to transport through the atmosphere above the surface receptors), quasi-laminar boundary layer ($R_b$, the resistance to transport across the thin layer of air that is in contact with the surface and varies with the diffusion of the pollutant transported), and surface resistances ($R_s$, the resistance to the uptake of the pollutant by the surface receptor, typically vegetation or soil)."

Page 4, Line 11: I am not sure whether the 110th meridian west is appropriate to divide east and west. There are more sites in the east than the west if 110ºW is used. A map with 110ºW and all the sites should be included in the supplement.
Response: We agree that the use of the 110$^{th}$ meridian to define the East vs West U.S. is somewhat arbitrary. It was used primarily because the majority of the $SO_2$ and $NO_x$ emissions in the U.S. are east of this meridian. We now include a map of the distribution of observation sites in the supplement. Please see supporting Fig. S1.

Page4, Line 13: How did the authors get the value of 0.984?

**Response:** 0.984 is the ration of molecular weight of $NO_3^-$ to the molecular wieght of $HNO_3$ (62/63) and is used to convert the mass of $HNO_3$ deposited to that of $NO_3^-$, as also previously discussed in Appel et al. (2011).

Page 5, ln 3.1, Could the authors be more specific about the improvements since Appel et al (2011)?

**Response:** There are numerous differences between the model configuration and versions used in this analysis and those previously used by Appel et al. CMAQv5.0 was used here and included the AERO6 aerosol module, while Appel et al. used CMAQv4.7 that employed the AERO5 aerosol module. Specific process differences between model versions can be found at: https://www.epa.gov/cmaq/cmaq-models-0. In the revised manuscript we point interested users to specifics of these model versions by including the following sentence after Pg 6 line 20: "There are numerous differences between the model configuration and versions used in this analysis and those previously used by Appel et al. (2011). Specific model process representation differences between CMAQv5.0 used here and CMAQv4.7 used in Appel et al. (2011) can be found at: https://www.epa.gov/cmaq/cmaq-models-0."

Page 5, It seems like the authors only did model evaluation and model justification for wet deposition. How was model performance for dry deposition? The authors could use data from AMON, IMPOROVE and EPA CASTNET to do this work.

**Response:** The reviewer raises an interesting point related to evaluation of dry deposition estimates. The U.S. CASTNET (Clean Air Status and Trends Network) did provide the dry deposition data. However, these values are not measured but instead derived using the inferential method, pairing the measured air pollutants concentration with a modeled deposition velocity from the MLM model (Meyers et al, 1998). So rather than comparing the two model values between CMAQ and MLM, we chose to compare CMAQ estimated ambient concentrations of both gaseous ($SO_2$) and particulate ($SO_4^{2-}$, $TNO_3^-$, $NH_4^+$) species with measurements from CASTNET. We change the title in section 2.2 "Wet deposition observations in the U.S." to "Deposition observations in the U.S.".

On Page 5 line 7, we add the description for the dry deposition from the U.S. CASTNET: "The U.S. CASTNET provides long-term observation of atmospheric concentrations as well as the dry deposition (https://www.epa.gov/castnet, accessed May 7, 2018). However, the dry deposition values reported are not directly measured, but estimated using the inferential method, pairing the measured air pollutant concentration with a modeled deposition velocity from the MLM model (Meyers et al, 1998). So rather than comparing dry deposition estimates from two models, we choose to evaluate the model's performance in simulating the ambient air concentrations (sulfur dioxide ($SO_2$), sulfate ($SO_4^{2-}$), total nitrate ($TNO_3 = NO_3^- + HNO_3$), and ammonium ($NH_4$)). The detailed site information and the number of years of observational data used for the model evaluation can be found in supporting Table S3. We apply the same criteria in selecting valid observation sites as the NADP/NTN."

In the revised manuscript we further discuss the evaluation of these air concentrations on Page 6 line 21:

"To evaluate the model's performance in simulating the DDEP, we compare the model simulated concentration with the observations from CASTNET. Comparisons of annual average simulated concentrations with corresponding measurements at the CASTNET sites show strong correlation for $SO_2$ (R of 0.88), $SO_4$ (0.95), $TNO_3$ (0.94), and $NH_4$ (0.94). Some underestimation for $SO_4$, and overestimation in other species ambient concentrations is noted (supporting Fig. S4). The model also captures the trends for these species with very high R, but the magnitude of the decreasing trends is underestimated by the model (supporting Fig. S5)."

Page 5, Line 19: Change "models" to "model results"
**Response:** We thank the reviewer for pointing this out. We now changed the word "model" to "model results".

Page 6, Line 15: It should be "Table S3"
**Response:** We now corrected from "supporting Table 3" to "supporting Table S3".

Page 6, Line 26 – Page 7, Line 2: Please explain the reasons for those results.
**Response:** The discrepancies for the trends of the TIN TDEP over U.S. ecoregion regions are caused by the combination of the decrease of the $NO_x$ emissions, and unregulated but increased $NH_3$ emissions at different places. We now clarify this on Page 6 line 26-Page 7 line 2:
"During the period from 1990 to 2010, TIN TDEP has significantly decreased (with p <0.05 for the standard two-tailed Student's t-test) over several ecoregions, including Eastern Temperate Forests, Northern Forests, Mediterranean California and Marine West Coast Forest (decreasing trend of 0.12, 0.071, 0.038 and 0.017 kg N ha$^{-1}$ yr$^{-1}$ respectively) as a result of significant reductions in anthropogenic $NO_x$ emissions (Xing et al., 2013)."

Page 7, Line 11- 20: Which one dominates the decrease of TSOx, $SO_4^{2-}$ or $SO_2$?
**Response:** We thank the reviewer for pointing this out. After performing additional calculations, we determined that the decrease of $TSO_X$ is dominated by $SO_4^{2-}$. We now add this information in Page 7 line 17:
"All the ecoregions experienced statistically significant decreases of TS TDEP over the past two decades which was dominated by the decreases in $SO_4^{2-}$, except for the Mediterranean California ecoregion which showed an insignificant decreasing trend (Table 6)."

Page 26, The legend of Fig 8 (a) needs to be fixed.
**Response:** We thank the reviewer for pointing this out. We have now fixed the legend on Fig. 8(a). Please see the updated plot in our revised manuscript.

---

## Author Comment (AC2) · 29 May 2018

Response to review #2 on acp-2018-116

Long-term trends in total inorganic nitrogen and sulfur deposition in the U.S. from 1990 to 2010

Yuqiang Zhang, Rohit Mathur, Jesse O. Bash, Christian Hogrefe, Jia Xing, Shawn J. Roselle

We thank referee #2 for the positive comments and constructive suggestions, which have helped us improve the manuscript. Summarized below are our detailed response to the reviewer comments (shown in blue). All comments have been carefully addressed here (blue colors are for referee's comments), and we have tracked all changes in the revised manuscript.

This paper examines trends in inorganic nitrogen and sulfur deposition from 1999 to 2010 across the U.S. This analysis is performed using WRF-CMAQ model simulations. The results from the model are compared to data from the NADP (National Atmospheric Deposition Program) Network. The trends and spatial patterns observed are discussed. Overall, this is a good paper.
Response: We thank the reviewer for the overall positive assessment of our paper.

But I do have some concerns. I feel a large part of the methods section is missing as the authors do not actually discuss the dry deposition data being used.
Response: We thank the reviewer for raising this issue. The U.S. CASTNET (Clean Air Status and Trends Network) does provide an estimate of long-term trends of dry deposition data. However, these values are not measured but instead derived using the inferential method, pairing the measured air pollutants concentration with a modeled deposition velocity from the MLM model (Meyers et al, 1998). So rather than comparing two model values from CMAQ and MLM, we chose to compare CMAQ outputs to ambient concentrations of both gaseous ($SO_2$) and particulate ($SO_4^{2-}$, $TNO_3^-$, $NH_4^+$) species with measurements from the CASTNET. We have added descriptions of the observation dataset from CASTNET and evaluation into manuscript, as also detailed in our response to a similar query by Reviewer 1. Please see section 2.2 and section 3.1.

We have also addressed the reviewer's similar comments in the specific comments below.

References:
Meyers, T. P., Finkelstein, P., Clarke, J. and Ellestad, T. G.: A multilayer model for inferring dry deposition using standard meteorological measurements, J. Geophys. Res. Atmos., 103(D17), 22645–22661, 1998.

General Comments: -I am a bit surprised that the abbreviation TSOx is used for sulfur deposition rather than TS. TS to me seems more fitting, but I understand if the other is more traditionally used as I am not as familiar with that literature as I am with nitrogen deposition. However, that being said it seems that the paper goes back and forth using TSOx and TSO4 to represent sulfur deposition. This is true throughout the main text, figures, and supporting information. This should be checked.
Response: We thank the reviewer for pointing this out and agree that consistency is needed in the use of the abbreviation. We agree with the reviewer that TS is a better abbreviation since total

sulfur deposition (expressed in mass of S) is analyzed and compared between the model and observations. In the revised manuscript we have replaced "TSO$_x$" and "TSO$_4$" in all the text, figures and tables as well as in the supporting information, with the abbreviation "TS".

Specific Comments: Abstract Page 1, Line 12 – The abbreviation WRF-CMAQ is not defined.
**Response:** In the revised manuscript, we have revised the text as:
"Here, we use long-term model simulations from the coupled Weather Research and Forecasting and the Community Multiscale Air Quality (WRF-CMAQ) model"

We also updated the sentence on line 16, Page 3:
"The long term simulations were previously performed using the coupled Weather Research and Forecasting and the Community Multiscale Air Quality (WRF-CMAQ model, Wong et al., 2012)"

Page 1, Lines 15-17 – The authors mention that the model generally underestimates the wet deposition. But they do not provide any reasons why this is. This should be added to the abstract.
**Response:** The underestimation of the wet deposition likely arises due to a combination of factors including coarse model grid resolution, missing emissions of lightning NO$_x$, as well as the poor temporal and spatial representation of NH$_3$ emissions. Now we add the explanation in Page 1 line 17:
"The underestimation of the wet deposition by the model is mainly caused by the coarse model grid resolution, missing lightning NO$_x$ emissions, as well as the poor temporal and spatial representation of NH$_3$ emissions."

Page 1, Line 19 – Suggest changing decrease of TNO$_3$ to decreases in TNO$_3$?
**Response:** Following the reviewer's suggestion, we have modified the sentence as:
"The decreasing trends of TIN TDEP are caused by decreases in TNO$_3$"

Page 1, Line 20 – The authors mention there are increasing trends in TIN deposition over the Tropical Wet Forest. This is the only region type listed in the text that does not have a geographic location included in its title. I think this makes it hard for readers to understand where it is. I would suggest adding a phrase such as southern Florida to aid the reader.
**Response:** We thank the reviewer for pointing this out. We have incorporated the reviewer's suggestions in Page 1 line 20:
"in the Tropical Wet Forests (Southern Florida Coastal Plain)"

Page 1, Line 22 – Suggest removing the words region of before Eastern
Page 1, Line 23 – Suggest removing the words region of before Tropical
**Response:** We followed the reviewer's comments and removed the words "region of" for the sentence from line 22-line 24. Now the new sentence is:
"TIN DDEP shows significant decreasing trends in the Eastern Temperate Forests, Northern Forests, Mediterranean California and Marine West Coast Forest, and significant increasing trends in the Tropical Wet Forests, Great Plains and Southern Semi-arid Highlands"

Page 1, Line 28 – Suggest adding an a before combination.
**Response:** We made the change following the reviewer's comments. The new sentence is:
"TDEP of TIN over the U.S. was dominated by deposition of $TNO_3$ during the first decade, which then shifts to reduced nitrogen ($NH_X$) dominance after 2003 resulting from a combination of NOx emission reductions and $NH_3$ emission increases."

1.Introduction Page 2, Line 12 – Suggest changing twice higher than to twice as high as
**Response:** We made change following the reviewer's comments. The new sentence is:
"Another possible source of $NH_3$ emissions are from vehicles which may be twice as high as the emission estimates in the current NEI (Sun et al., 2016)."

Page 2, Line 13 – Suggest removing the the before sulfur.
Page 2, Line 14 – form fossil-fuel should be from fossil-fuel
**Response:** We removed word "the" in the sentence, and also corrected the word "form". The new sentence is:
"The primary emission source for sulfur deposition is sulfur dioxide ($SO_2$) which also mainly originates from fossil-fuel combustion (Smith et al., 2011)"

Page 2, Lines 15-20 – Here the authors discuss the wet deposition national networks. But they do not actually tell how the measurements are made. I would suggest adding some text telling how the samples are collected and then measured by ion chromatography to provide the data.
**Response:** Following the reviewer's suggestion in the revised manuscript we include a brief description of the NADP measurements in section 2.2 (Page 4 line 6):
"The deposition is measured by wet-only samples, which are triggered by precipitation. The deposition of sulfate and nitrate are analyzed by ion chromatography, and ammonium by flow injection analysis (http://nadp.slh.wisc.edu/educ/sample.aspx, accessed May 4, 2018)."

Page 2, Line 21 – Suggest adding the words e.g., before EEA. Also a comma is missing after EEA
Page 2, Line 22 – Suggest removing the comma and phrase to name a few after 2015
**Response:** We have incorporated the reviewer's suggestion by modifying the sentence as:
"These data have been extensively used to quantity the sources, pattern, and temporal trends of WDEP of major species worldwide (e.g., EEA, 2011; Jia et al., 2014; Cheng and Zhang, 2017; Lajtha and Jones, 2013; Du et al., 2014; Sickles II and Shadwick, 2007a, 2007b, 2015)."

Page 3, Line 2 – There is an extra period after loss
**Response:** We thank the reviewer for catching the typo. The extra period has been removed in the revised text.

Page 3, Line 5 – Suggest removing the second Zhao et al.
**Response:** Actually, the first Zhao et al., 2009, and the second Zhao et al., 2015, 2017 are not the same first author, even though they share the same last name and initials. To made it clear, now we rewrite this:
"Zhao Y. et al., 2009; Zhao Y. H. et al., 2015, 2017"

Page 3, Line 8 – Suggest changing process to processes

**Response:** We made the change following the reviewer's comments. The new sentence now is:
"CTMs can link the sources to the deposition through atmospheric chemistry and transport processes"

2.Methods 2.1.Model setup Page 3, Line 16 – The abbreviation WRF-CMAQ is not defined

**Response: We** thank the reviewer for pointing this out. Now we add the abbreviation:
"The long term simulations were previously performed using the coupled Weather Research and Forecasting and the Community Multiscale Air Quality (WRF-CMAQ model, Wong et al., 2012)"

Page 3, Line 20 – There is an extra comma after Gan et al. Also suggest adding an a before detailed

**Response:** We made the change following the reviewer's comments. The new sentence now is:
"Interested readers are referred to Gan et al. (2015, 2016) for a detailed description of the settings of the CMAQ model and physical configurations of the WRF model (Table S1)."

Page 3, Line 22 – The chemical abbreviation used are not defined

**Response:** In the revised manuscript we have added the chemical abbreviation as:
"The performance of the coupled WRF-CMAQ model for major trace gases, aerosol species and meteorological variables such as ozone ($O_3$), fine particulate matter ($PM_{2.5}$) and aerosol optical depth at both the hemispheric and regional scale …"

Page 3, Line 24 – Suggest changing was shown to has shown

**Response:** We have corrected this as:
"and has shown skill in simulating the magnitudes and long-term trends of these variables."

2.2.Wet deposition observations in the U.S. Page 4, Line 11 – Suggest changing observation data used for to observational data used for the

**Response:** We modified the sentence following the reviewer's suggestion as:
"The detailed site information and the number of years of observational data used for the model evaluation"

Page 4, Line 13 – Suggest changing combine WDEP to combines WDEP and with 0.984 to by 0.984

**Response:** We have incorporated the reviewer's suggestion in the revised sentence as:
"In pairing the observed and modeled $TNO_3$ WDEP values (which combines WDEP of $NO_3^-$ and $HNO_3$), we multiply the model estimated $HNO_3$ WDEP by 0.984 to account for the transformation of $HNO_3$ to $NO_3^-$ in solution in the measurements"

Page 4, Line 15 – Suggest changing combine WDEP to combines WDEP and with 1.06 to by 1.06

**Response:** We have incorporated the reviewer's suggestion in the revised sentence as:
"In pairing the observed and modeled $NH_X$ WDEP values (which combines WDEP of $NH_4^+$ and $NH_3$), we multiply the model estimated $NH_3$ WDEP by 1.06"

**Response:** We have incorporated the reviewer's suggestion in the revised sentence as:
"In pairing the observed and modeled $TSO_X$ WDEP values (which combines WDEP of $SO_4^{2-}$ and $SO_2$), we multiply the model estimated $SO_2$ WDEP by 1.50"

**Response:** The reviewer raises an interesting point related to evaluation of dry deposition estimates. The U.S. CASTNET (Clean Air Status and Trends Network) did provide the dry deposition data. However, these values are not measured but instead derived using the inferential method, pairing the measured air pollutants concentration with a modeled deposition velocity from the MLM model (Meyers et al, 1998). So rather than comparing the two model values between CMAQ and MLM, we chose to compare CMAQ estimated ambient concentrations of both gaseous ($SO_2$) and particulate ($SO_4^{2-}$, $TNO_3^-$, $NH_4^+$) species with measurements from CASTNET. We change the title in section 2.2 "Wet deposition observations in the U.S." to "Deposition observations in the U.S.".

On Page 5 line 7, we add the description for the dry deposition from the U.S. CASTNET:
"The U.S. CASTNET provides long-term observation of atmospheric concentrations as well as the dry deposition (https://www.epa.gov/castnet, accessed May 7, 2018). However, the dry deposition values reported are not directly measured, but estimated using the inferential method, pairing the measured air pollutant concentration with a modeled deposition velocity from the MLM model (Meyers et al, 1998). So rather than comparing dry deposition estimates from two models, we choose to evaluate the model's performance in simulating the ambient air concentrations (sulfur dioxide ($SO_2$), sulfate ($SO_4^{2-}$), total nitrate ($TNO_3 = NO_3^- + HNO_3$), and ammonium ($NH_4$)). The detailed site information and the number of years of observational data used for the model evaluation can be found in supporting Table S3. We apply the same criteria in selecting valid observation sites as the NADP/NTN."

In the revised manuscript we further discuss the evaluation of these air concentrations on Page 6 line 21:
"To evaluate the model's performance in simulating the DDEP, we compare the model simulated concentration with the observations from CASTNET. Comparisons of annual average simulated concentrations with corresponding measurements at the CASTNET sites show strong correlation for $SO_2$ (R of 0.88), $SO_4$ (0.95), $TNO_3$ (0.94), and $NH_4$ (0.94). Some underestimation for $SO_4$, and overestimation in other species ambient concentrations is noted (supporting Fig. S4). The model also captures the trends for these species with very high R, but the magnitude of the decreasing trends is underestimated by the model (supporting Fig. S5)."

Page 4, Lines 2-26 – The authors do not actually explain how the data from the network is collected. I understand the model analysis is the point of the paper. But since these

observational data are used to evaluate the model then the authors should provide at
least some text to give the readers context.
**Response:** We have now included a brief description of the NADP measurement in Page 4 line 6:
"The wet deposition is measured by wet-only samples, which are triggered by precipitation. The
deposition of for sulfate, nitrate are analyzed by ion chromatography, and ammonium by flow
injection analysis (http://nadp.slh.wisc.edu/educ/sample.aspx, accessed May 4, 2018)."

Page 5, Line 5 – There is an extra comma after equation 2
**Response:** We thank the reviewer for pointing this out. Now we removed the extra comma.

Page 3, Line 14 to Page 5, Line 6 – In the methods section there is no discussion of
the trend analysis that is used throughout the paper. What is this analysis? How is it
done? This should be added to the paper.
**Response:** Thanks the reviewer for pointing this out. We now added the descriptions how we
performed the trends analysis in Page 4 line 26:
"For the trend analysis, we focus on the linear trends (Colette et al., 2011; Xing et al., 2015a), in
which the linear least square fit method is employed, and significance of trends is examined with
a Student t-test at the 95% confidence level (p=0.05)"

References:
Colette, A., Granier, C., Hodnebrog, Ø., Jakobs, H., Maurizi, A., Nyiri, A., Bessagnet, B.,
D'Angiola, A., D'Isidoro, M., Gauss, M., Meleux, F., Memmesheimer, M., Mieville, A., Rouïl, L.,
Russo, F., Solberg, S., Stordal, F., and Tampieri, F.: Air quality trends in Europe over the past
decade: a first multi-model assessment, Atmos. Chem. Phys., 11, 11657–11678, doi:10.5194/acp-
11-11657-2011, 2011.
Xing, J., Mathur, R., Pleim, J., Hogrefe, C., Gan, C.-M., Wong, D. C., Wei, C., Gilliam, R., and
Pouliot, G.: Observations and modeling of air quality trends over 1990–2010 across the Northern
Hemisphere: China, the United States and Europe, Atmos. Chem. Phys., 15, 2723-2747,
doi:10.5194/acp-15-2723-2015, 2015a.

3.Results 3.1.Model evaluation of WDEP
Page 5, Line 16 – Suggest changing increases for all the three to increase for all three. Also exhibit
should be exhibited. Also suggest changing in east than that in west to in the east than the west
**Response:** We followed the reviewer's comments, and have rewritten the sentence:
"After performing the precipitation adjustment, the NMB values increase for all three species
(Table 1). The model exhibited better performance for WDEP in the east than the west"

Page 5, Line 17 – A period is missing after (Appel et al., 2011)
**Response:** We thank the reviewer for pointing this out and have corrected it in the revised
manuscript.

Page 5, Line 19 – Suggest changing both observations and models to both the observations
and model results
**Response:** We thank the reviewer for pointing this out. Now we have corrected this:
"as seen from both the observations and model results (Table 2)"

Page 5, Line 20 – Suggest adding a the before Tropical

**Response:** Now we added the word "the" before Tropical.

Page 6, Line 11 – I am not sure I understand the phrase but a slightly distinctions in trends for different ecoregions. Is it maybe but with slight distinctions in the trends for each ecoregion?

**Response:** We now rewrite the sentence as the reviewer suggested:

"and the model is also able to capture these very well but a slightly distinctions in the trends for each ecoregion"

Page 6, Lines 11-13 – The authors mentions that the model generally underestimates decreasing WDEP trends for all sites, but for NH$_x$ they see increasing WDEP trends. Why is this? The authors need to tell why they think this might be the case for the model.

**Response:** The magnitude of the decreasing trends in TNO$_3$ and TS wet deposition are slightly underestimated by the model and result from both the coarse model grid resolution and uncertainties in the emission inventories. We have add this in Page 6 line 12:

"We see that the model generally underestimates the magnitude of the decreasing WDEP trends at many sites for TNO$_3$ and TS (Tables 2 and 4), which may be caused by the coarse model grid resolution (36km), and uncertainties in the emission inventories."

Page 6, Line 14 – Suggest removing the word results before model

**Response:** We removed the repeat "results" as suggested:

"our model results indicate larger bias"

Page 6, Line 15 – Suggest changing increases for all the three to increase for all three. Also why are the authors only looking at the data from 2002-2006 when they discuss the NMB increase observed? This needs to be clarified.

**Response:** We now modified the sentence based on the reviewer's comments:

"The NMB increase for all three species in our results from 2002 to 2006"

The reason why we looked at the data from 2002-2006 only as Appel et al. (2011) only has the simulations from 2002 to 2006. Here we want to compare the performance between the model runs using a newer version of the CMAQ model which was used in our study, with older version of the CMAQ from previous study.

Page 6, Line 18 – Suggest changing are more to have more

Page 6, Line 19 – Suggest changing challenging to challenges

**Response:** We made the changes following the reviewer's these two comments:

"coarse resolution models (e.g. 36km in our study) have more challenges to simulate"

Page 5, Line 8 to Page 6, Line 20 – Why is there no matching section on the model evaluation for DDEP? The remainder of the results section discusses the trends in total, wet, and dry deposition so it seems that it should be established how the model compares with the calculated dry deposition values provided by NADP.

**Response:** We thank the reviewer for pointing this out. Please see our reply above for the similar question.

3.2.Spatial patterns of modelled total deposition of nitrogen and sulfur Page 6, Line 21
– modelled should be written as modeled to be consistent with how it is used throughout
the rest of the text

**Response:** We thank the reviewer for pointing this out. We now kept consistent with the words we used, and made the following changes:

Page 6 line 21: "Spatial patterns of modeled total deposition of nitrogen and sulfur"

Page 5 line 2-3: "account for biases in modeled precipitation by adjusting the modeled WDEP"

Page 7, Line 18 – Suggest removing the and after showed
Page 7, Line 19 – Believe that Table 4 should be Table 6

**Response:** We reply the reviewers' above two comments together. We now removed the word "and" after the word "showed", and change "Table 4" to "Table 6". The new sentence is: "which showed insignificant decreasing trend (Table 6)".

3.3.Wet versus dry nitrogen and sulfur deposition trends in the U.S.
Page 7, Line 25 –Suggest adding a the before Eastern
Page 7, Line 26 – Suggest adding a the before Northern and Great

**Response:** We thank the reviewer for pointing this out. We reply the reviewers' above two comments together. Now we have add the word "the" as the reviewer suggested:

"The most significant decreasing region is the Eastern Temperate Forests, with an annual decrease of -0.070 kg N ha$^{-1}$ yr$^{-1}$, followed by the Northern Forests (-0.037 kg N ha$^{-1}$ yr$^{-1}$) and the Great Plains (-0.023 kg N ha$^{-1}$ yr$^{-1}$)"

Page 7, Line 27 – Suggest changing was mainly to were mainly

**Response:** We thank the reviewer for pointing this out. Now we have corrected this. "The decreasing trends of TIN WDEP were mainly caused"

Page 7, Line 28 – The authors mention that there are no significant changes for WDEP
of NHx. However, in Table S4 the values for Tropical Wet Forests are in bold, which is
what indicates a significant trend. Also there is light blue being shown in Figure S4b.
This needs to be clarified.

**Response:** We thank the reviewer for pointing this out. There are actually no significant trends for WDEP of $NH_X$ in the majority of U.S. Now we have rewrote this sentence:

"There are no significant changes for WDEP of $NH_X$ in the majority of U.S. except for the region Tropical Wet Forests (supporting Fig. S4b),"

Page 8, Line 8 – Suggest adding an a before distinct and changing value to values

**Response:** We have made the changes following the reviewer's comments:

"Fig. 7 shows a distinct spatial distribution for both the WDEP and DDEP of sulfur, with much higher values in the eastern U.S."

Page 8, Line 9 – Suggest adding a the before vicinity and changing source to sources

**Response:** We have made the changes following the reviewer's comments:

"in the vicinity and downwind of major sources"

3.4.Deposition budget in U.S.
Page 8, Line 18 – Suggest changing were estimated to was estimated

**Response:** We modified the sentence following the reviewer's comment:
"The TNO$_3$ WDEP was estimated to decrease"

Page 8, Line 19 – Suggest removing the hyphen after 2010
**Response:** We removed the hyphen as suggested.

Page 8, Line 21 – Suggest changing changes to changed
**Response:** Changed as suggested:

Page 8, Line 22 – Suggest changing till to until and removing the the before NHx
**Response:** We thank the reviewer for pointing this out. Now we have modified the sentence:
"TNO$_3$ deposition dominates TIN TDEP until the early 2000s. After 2003, however, NHx dominates the TIN TDEP over the U.S."

Page 8, Line 26 – Suggest changing 1999-2010 to 1999 to 2010
**Response:** We have changed "1990-2010" to "1990 to 2010":

Page 8, Line 27 – Suggest changing emission to emissions
**Response:** We made the change following the reviewer's comment:
"due to regulations and growth in NH$_3$ emissions"

Page 8, Line 28 – The reference is written in blue
**Response:** We have reformatted the reference.

Page 9, Line 2 - The references are written in blue
**Response:** We have reformatted the reference.

Page 9, Lines 1-5 – I believe that this section is in reference to Figure 8, but there is citation to Figure 8 listed here.
**Response:** We now add the reference to Figure 8 in the Page 9 line 1:
"Similar to TIN TDEP, the TSO$_X$ TDEP has also decreased, from 6.85 kg S ha$^{-1}$ yr$^{-1}$ in 1990 to 3.26 kg S ha$^{-1}$ yr$^{-1}$ in 2010 (Fig. 8 (b)),"

Conclusions Page 9, Line 10 – Suggest changing observation to observations
**Response:** We made the change as suggested.

Page 9, Line 25 – Suggest adding a the before Eastern
**Response:** We add the word "the" as suggested.

Page 10, Line 9 – It should be aerosol-phase
**Response:** We change the word "aerosol phase" to "aerosol-phase".

Data availability Page 10, Line 18 – Suggest changing shared to obtained
**Response:** We changed the word "shared" to "obtained" as the reviewer suggested.

Competing interests Page 10, Line 21 – Suggest changing conflict to conflicts

**Response:** We changed "no conflict of interest" to "no conflicts of interest".

Page 10, Line 26 – Suggest adding a the before U.S. and removing the phrase improvements of after suggestions on the
**Response:** We made the changes following the reviewer's comments, and the new sentence is: "We greatly acknowledge James Kelly and Kristen Foley from the U.S. EPA for their comments and suggestions on the initial version of this manuscript."

Disclaimer Page 10, Line 28 – Suggest changing view to views
**Response:** We changed the word "view" to "views".

References Page 11, Line 26 – Believe the accent marks in Muller should be over the u
**Response:** Now we changed to "Müller" as the reviewer suggested.

Tables and Figures
Table 1 -In first line of caption – Suggest changing for all the annual to for the sum of the annual -In second line of caption - Suggest adding a the before model -What is the difference between R and R for trends? There is no discussion about this in the main text so it is hard to understand why the two set of values are being shown.
**Response:** We have modified the sentence following the reviewer's comments and also explain what the second R means. The "R for the trends" are the correlation coefficient for the 21-yr changes of the wet deposition ($TNO_3$, $NH_X$ and $TSO_x$) between the model and the observations Now the new caption is:
"Correlation coefficient (R), mean bias (MB, kg ha$^{-1}$), and normalized mean bias (NMB, %) for the sum of the annual accumulated wet deposition (WDEP) between the model and NADP sites from 1990 to 2010, including both the model values with and without applying monthly/annual precipitation adjustment. The R for trends are the correlation coefficient for the 21-yr changes of the wet deposition ($TNO_3$, $NH_X$ and TS) between the model and the observations."

Table 2 -In first line of caption – Suggest adding a the before 10 -In third line of caption – There should be a hyphen in t-test -Second column heading – Suggest changing Regions to Region -Third column heading – Suggest changing # sites to # of sites
**Response:** We changed the caption as the reviewer's suggested, and also made the changes to the Table. Please see our new draft.

Table 3 -Second column heading – Suggest changing Regions to Region -Third column heading – Suggest changing # sites to # of sites
Table 4 -Second column heading – Suggest changing Regions to Region -Third column heading – Suggest changing # sites to # of sites
Table 5 -Second column heading – Suggest changing Regions to Region
Table 6 -In third line of caption – There should be a hyphen in t-test -Second column heading – Suggest changing Regions to Region
**Response:** We answer the reviewers' above comments about Table 3 to Table 6 together. As suggested, we have made the changes in the tables' captions. Please see our new draft.

Figure 1 -In second line of caption – To match the figure between observations and precipitation-adjusted model results should be switched -In third line of caption – Suggest changing Each NADP to The data at each NADP site -Letters should be added to each plot and the caption updated to indicate this -Suggest making a symbol indicating that green is for East sites and red is for West sites as currently this is only indicated from the small text at the top of each plot -It should be indicated in the caption what the solid and dashed lines in each plot represent -There are no subscripts in the abbreviations used on both the x and y-axes for all plots

**Response:** Following the review's comments, we have made the following changes. The new captions reads as:

"Scatter plots for the annual accumulated WDEP (total oxidized nitrogen ($TNO_3$, a), reduced nitrogen ($NH_X$, b), and total sulfate (TS, c)) between precipitation-adjusted model results and observations from 1990 to 2010 for 170 valid sites with 3531 valid data points. The data at each NADP is assumed to be valid for our analysis only if at least 18 years of observation data are available at that site and the data coverage is at least 75% for each year. Each point in the plots represents the annual accumulated WDEP for a given site and year. Note that the annual accumulated WDEP values used in this analysis may not be the actual annual totals due to missing data in the observations. The green color is for the eastern U.S., and the red color is for the western U.S., with the dashed line for the 1:2 and 2:1 ratio, and the solid line for the 1:1 ratio."

We also add subscripts for abbreviations at both the x, y-axes of all the plots. Please see our updated figures.

Figure 2 -In first line of caption – Suggest changing of (a) TNO3 to for (a) TNO3 - In second line of caption – Suggest removing the phrase annual accumulated before precipitation. -In second, third, and fourth lines of caption - US should be U.S. -There are no x-axis labels -The legend for plots a, b, and c are incorrect as they indicate the data for the East is red and West is green

**Response:** We have corrected the legends for plots a, b, and c. Please see our updated draft.

Figure 3 -In first line of caption – Suggest changing adding a the before observations -In first line of caption – To match the figure between observations and precipitationadjusted model valves should be switched -In second line of caption – Suggest changing observation to observational -Letters should be added to each plot and the caption updated to indicate this -It should be indicated in the caption what the solid and dashed lines in each plot represent -There are also no subscripts in the abbreviations used on both the x and y-axes for all plots

**Response:** We have made the changes to the captions, and also added the letters for the plots. Please see the new plots from our updated draft.
The new caption is:
"Figure 3. Comparison of the WDEP trend for each valid site between the precipitation-adjusted model values and observational for total oxidized nitrogen ($TNO_3$, a), reduced nitrogen ($NH_X$, b), and total sulfate (TS, c). Each NADP site is assumed to be valid for our analysis only if at least 18 years of observation data are available at that site and the data coverage is at least 75% for

each year. The green color is for the eastern U.S., and the red color is for the western U.S., with the dashed line for the 1:2 and 2:1 ratio, and the solid line for the 1:1 ratio. "

Figure 4 -In first and second lines of caption – Suggest changing panel to panels -In third line of caption – Suggest changing plot show p value to plots show p values -In fourth line of caption – Suggest adding a comma after i.e. -There are no x and y-axes labels

**Response:** We have made changes to the captions according to the reviewer' comments.

Figure 5 -In second line of caption – Suggest changing the right plot show p value great than to both plots show p values greater than -In third line of caption – T-test should be t-test. Also suggest adding a comma after i.e. -Letters should be added to each plot and the caption updated to indicate this -There are no x and y-axes labels

Figure 6 -In first line of caption – Suggest changing (top panel) and DEP (bottom panel) to (top panels) and DDEP (bottom panels) -In second line of caption – Suggest changing plot show p value great than to plots show p values greater than -In third line of caption – T-test should be t-test. Also suggest adding a comma after i.e. -There are no x and y-axes labels

**Response:** We have now corrected this. Please see our updated draft for the new plots.

Figure 8 -In caption – It should be mentioned in the caption that the percent contribution is being indicated on each bar -In first line of caption – US should be U.S. -On the yaxis for both plots, US should be U.S. -Suggest in legend for plot a calling Oxid as NO3 instead and Red as NHx instead so that it matches the main text

**Response:** We now mention the percentiles in the caption, and also updated our plots.

"Interannual variability of the TDEP for inorganic nitrogen (a), and sulfur (b) in the U.S. from 1990 to 2010, including their fractions labelled as percent contributions for WDEP of oxidized nitrogen ($NO_3$), WDEP of reduced nitrogen ($NH_X$), DDEP of oxidized nitrogen ($NO_3$) and DDEP of reduced nitrogen ($NH_X$) deposition for the nitrogen, and WDEP versus DDEP for sulfur. "

Figure 9 -In second line of caption – Suggest changing an NHx to a NHx -In third line of caption – Suggest removing the comma after 0.5 -There are no x and y-axes labels -Title for plot a – Suggest changing NHx ratio over TIN 1990 to TDEP NHx to TIN ratio 1990 -Title for plot b – Suggest changing NHx ratio over TIN 2010 to TDEP NHx to TIN ratio 2010 -Title for plot c – I am not sure I understand this plot title. What is (/year) indicating? Should the title maybe be TDEP NHx to TIN ratio Overall Trend?

**Response:** We have made the changes suggested by the reviewer in the revised manuscript. Please see our new draft for the updated plots.

Supporting Information Figure S1 -In caption – Suggest changing all the ofs to equal signs (e.g., 5 of Northern Forests to 5 = Northern Forests) -There are no x and y-axes labels -In plot title – US should be U.S. Also what does mask mean? It is not indicated in the caption or text.

**Response:** We have made the changes following the reviewer's comments, and also update the plot caption as "U.S. ecoregion Level 1". Please see the plot in our updated draft. The new caption is:

"The 10 Level I ecoregions in the continental U.S.: 5 = Northern Forests, 6 = Northwestern Forested Mountains, 7 = Marine West Coast Forest, 8 = Eastern Temperate Forests, 9 = Great Plains, 10 = North American Deserts, 11 = Mediterranean California, 12 =Southern Semi-arid Highlands, 13 = Temperate Sierras, and 15 = Tropical Wet Forests."

Figure S2 -In first line of caption – Suggest changing plot to plots. Also the words observation and model should be switched to match what is actually plotted. -In second line of caption – Suggest changing data. The site in NADP is assumed to data points. The data at each NADP site is assumed -In third line of caption – Suggest changing valid if only at to valid only if at, changing is available to are available, and changing for the to for that -In fourth line of caption – suggest changing plot to plots -In fifth line of caption – Suggest removing the the before missing -In sixth line of caption – Suggest changing observation to observations -It should be indicated in the caption what the solid lines in the plot represent. Also should this be like the other plots and have two dashed lines and one solid line?

**Response:** We made the changes following the reviewer's comments.

Figure S3 -Letters should be added to each plot and the caption updated to indicate this -Suggest making a symbol indicating that green is for East sites and red is for West sites as currently this is only indicated from the small text at the top of each plot -It should be indicated in the caption what the solid and dashed lines in each plot represent -There are no subscripts in the abbreviations used on both the x and y-axes for all plots

**Response:** We add letters for each plot, and update the captions to indicate this. We also add descriptions for the two dashed and solid lines. The new captions is:

"Scatter plots for the annual accumulated deposition (total oxidized nitrogen ($TNO_3$, a), reduced nitrogen ($NH_X$, b), and total sulfate (TS,c)) without considering the precipitation adjustment between observation and model results from 1990 to 2010 for 170 valid sites with 3531 valid data. The site in NADP is assumed valid if only at least 18 years of observation data is available with 75% annual coverage for the site. Note that the annual accumulated deposition may not be the actual annual totals because of the missing data in the observation. The green color is for the eastern U.S., and the red color is for the western U.S., with the dashed line for the 1:2 and 2:1 ratio, and the solid line for the 1:1 ratio."

Figure S4 -There are no x and y-axes labels -There are no subscriptions in the abbreviations used in the titles for all plots

**Response:** We now add the subscriptions for all the abbreviations. Please see the new plots in our updated draft.

Figure S5 -In first line of caption – Suggest adding a the before US. Also US should be U.S. -There are no x-axis labels -Suggest changing y-axis labels to Fraction of the Total -Suggest pointing out on both plots somehow 2003 since this is an important year in terms of trends and so that it corresponds with the discussion in the main text. Maybe add a vertical dashed line.

**Response:** We add a red arrow to point the year 2003, and description in the caption. "The red arrow points to the year 2003."

Table S1 -Either the comma or the parenthesis should be removed from Xing et al. reference. Both are not needed.
**Response:** We removed the comma for both the Xing et al. (2013) and Xing et al. (2015a).

Table S3 -In third line of caption – Suggest removing the and with at the end of the sentence
**Response:** We removed the "and with" as suggested. Please see our new draft.

Table S4 -In fourth line of caption - There should be a hyphen in t-test -Second column heading – Suggest changing Regions to Region
**Response:** We made the changes as the reviewer suggested. Please see our new draft.

Table S5 -Second column heading – Suggest changing Regions to Region
**Response:** We made the change as the reviewer suggested. Please see our new draft.